# Construction and Preclinical Evaluation of a Recombinant Attenuated Measles Vaccine Candidate of the H1a Genotype

**DOI:** 10.3390/vaccines13060571

**Published:** 2025-05-27

**Authors:** Lixia Xie, Yuanbao Liu, Yajing Zhang, Biao Niu, Hui Wang, Yue Guo, Jinliang Wang, Juncheng Ruan, Guandong Xie, Zhiguo Wang, Zhenfang Fu, Qi An, Dayong Tian

**Affiliations:** 1R&D Department, Shanghai King-Cell Biotechnology Co., Ltd., Shanghai 201506, China; xielixia@king-cell.com (L.X.); zhangyajing@king-cell.com (Y.Z.); niubiao@king-cell.com (B.N.); wanghui@king-cell.com (H.W.); guoyue@king-cell.com (Y.G.); wangjinliang@king-cell.com (J.W.); ruanjuncheng@king-cell.com (J.R.); xieguandong@king-cell.com (G.X.); fuzhenfang@king-cell.com (Z.F.); 2Department of Expanded Program on Immunization, Jiangsu Provincial Center for Disease Control and Prevention, Nanjing 210009, China; liuyb@jscdc.cn (Y.L.); njwang1718@163.com (Z.W.)

**Keywords:** measles, genotype, recombinant virus, cross-protection, attenuated live

## Abstract

**Background**: Measles, an acute respiratory infectious disease caused by the measles virus, continues to pose a significant threat to children under five years old worldwide. Despite the availability of effective vaccines, challenges such as insufficient vaccination coverage and antigenic drift contribute to its persistence. Based on a newly isolated wild-type measles virus strain (genotype H1a), designated MVs/Jiangsu.CHN/38.16/1[H1a] (MV-1), this study aims to develop and evaluate a novel recombinant measles virus vaccine candidate designed to enhance immunogenicity and broaden protection against multiple epidemic genotypes. **Methods**: A recombinant measles virus vaccine candidate, designated rSchwarz/FH(H1a), was developed by incorporating immunogenic genes from the H1a genotype into the backbone of the Schwarz vaccine strain. The genetic stability, safety, and immunogenicity of this vaccine candidate were evaluated in preclinical models. Relevant sample sizes and methodologies were selected to ensure comprehensive assessment of vaccine efficacy against various genotypes (H1a, B3, D8). **Results**: The rSchwarz/FH(H1a) vaccine candidate demonstrated enhanced immunogenicity, with robust immune responses observed against the targeted genotypes. Additionally, it showed excellent genetic stability and safety profiles, indicating potential for effective use in vaccination programs. Notably, the vaccine provided cross-protection against multiple epidemic genotypes, highlighting its broader application in controlling measles outbreaks. **Conclusions**: Our findings suggest that the rSchwarz/FH(H1a) vaccine candidate represents a promising advancement in measles vaccine development. It has the potential to strengthen current measles vaccination strategies by providing improved immunogenicity and broader protection against different circulating genotypes. Further clinical trials are warranted to confirm these promising preclinical results.

## 1. Introduction

Measles, an acute respiratory infectious disease caused by the infection of measles virus, constitutes a substantial threat to the health and survival of children under five years of age [1]. Despite the existence of only one serotype of measles virus, it is classified into eight genotypes (A–H) and further subdivided into 24 genetic subtypes, based on sequence variations found in the hemagglutinin (H) or nucleoprotein (N) genes [2,3,4]. Infection with this virus causes skin rashes and often progresses to severe complications affecting the respiratory system, heart, or nervous system. Measles, caused by a single serotype of the measles virus with 24 genetic subtypes, was once a leading cause of child mortality, with 2.6 million deaths annually before the introduction of vaccines in 1963 [5,6].

Currently, measles vaccines, derived from attenuated genotype A strains, demonstrate robust immunogenicity and elicit both cellular and humoral immune responses against wild-type measles virus. Notably, vaccine strains such as Edmonston, Schwarz, AIK-C, and S191 are widely utilized globally [7]. Furthermore, these strains have served as viral vectors for the development of recombinant viral vaccines targeting various pathogens, including COVID-19, mumps, influenza, and HIV, thereby highlighting their potential for combined vaccine strategies [8,9,10,11,12,13,14]. Additionally, research has ventured into innovative immunization techniques, such as microneedles and inhalation immunization, aimed at enhancing the efficacy and convenience of measles vaccination [15,16].

However, the global measles epidemic has exhibited fluctuations over the past two decades, characterized by sporadic outbreaks in localized areas [17,18]. In 2022, the number of measles cases surged to approximately 9 million, causing roughly 136,000 deaths—marking a 18% increase in cases and a 43% increase in deaths compared to the previous year [19]. A substantial portion of these cases occurred in children under five, leading to severe consequences. The primary factor contributing to this resurgence is insufficient vaccination coverage. Considering the measles virus high R_0_ value, which ranges from 12 to 18, it is imperative to maintain an immunization rate above 70% in the long term to establish an effective protective barrier and prevent widespread outbreaks [20,21]. Factors, including the COVID-19 pandemic, have led to an estimated 22 million children worldwide missing their first dose of the measles vaccine in 2022, with an additional 11 million children missing their second dose [22]. Furthermore, the antigenic drift of the measles virus is a crucial factor that contributes to the recurrence of measles epidemics [23]. While studies have shown cross-protection among various measles virus genotypes, there is also evidence of different cross-protection [23,24,25]. Currently, genotypes H1, B3, and D8 predominate and exhibit distinct geographical patterns [26]. For example, the H1a genotype has been the most prevalent in China for nearly two decades, whereas B3 and D8 genotypes are widespread in Europe. Therefore, the development of novel measles vaccines that target these prevalent genotypes is crucial for addressing the threat that measles poses to human health.

In this study, we successfully constructed a recombinant measles virus based on the Schwarz strain, wherein its major immunogenic genes were replaced with corresponding sequences from the H1a genotype measles virus. This recombinant virus demonstrated potential as an attenuated live vaccine candidate, exhibiting superior replication and genetic stability, enhanced immunogenicity against measles virus of H1a, B3, and D8 genotypes, and high attenuation characteristics. These attributes render it a promising vaccine candidate against the currently circulating measles virus strains.

## 2. Materials and Methods

### 2.1. Cells, Viruses and Animals

BSR-T7 cells and measles virus Schwarz strain were obtained from King-cell Biotech. Vero cells were purchased from the American Type Culture Collection (ATCC). The Vero/hSLAM cells (ECACC 04091501) were developed by Cyagen Biosciences Inc. (Guangzhou, China), based on the Vero cells originally acquired from ATCC. Chicken embryo fibroblasts (CEF) were isolated from 9- to 11-day-old SPF chicken embryos obtained from Vital River Labs (Beijing, China) and cultured in medium supplemented with 0.2% hydrolyzed lactalbumin and 5% FBS.

A total of 21 measles virus strains of the H1a genotype were isolated from diverse regions across China, encompassing Beijing, Shaanxi, Guangdong, Jiangsu, Zhejiang, and Hubei. The strains D8 and MV-1, isolated by the Jiangsu Provincial Center for Disease Control and Prevention (CDC) from clinical samples, were passaged twice before experimental use. The Schwarz vaccine strain, purchased from the University of Georgia in the United States, was passaged twice in our laboratory before use in experiments. S191, a national reference strain for live attenuated measles vaccine, was obtained from the National Institutes for Food and Drug Control in China. The rSchwarz/FH(B3) is a recombinant virus strain constructed by King-cell Biotech, using the Schwarz strain as a backbone but replacing the F and H protein genes which were from a genotype B3 measles virus (GenBank accession number: KY969477.1). After the recombinant virus were successfully rescued, the virus was continuously passaged three times in Vero/hSLAM cells before use in experiments. The rAAV-hsyn-SLAM virus was prepared by King-cell Biotech.

All experimental animals, including mice and guinea pigs, were obtained from Vital River Laboratories, while IFNα/βR^−/−^ (A129) mice were provided by King-cell Biotech. The study was conducted in compliance with GLP standards and approved by the IACUC. Upon arrival, animals were housed in individually ventilated cages (IVCs) under controlled environmental conditions (temperature, humidity, 12 h light/dark cycle) and acclimated for 7 days prior to immunization. Animals were randomly assigned to groups using Microsoft Excel’s RAND() function, and each received a unique ID to ensure blinding throughout the study. Experimental procedures were performed in a fixed order to minimize variability, and control and experimental groups were housed separately to prevent cross-contamination. Control groups were always handled first during all procedures. Animals were monitored daily for signs of distress or adverse effects, and any animal exhibiting health issues unrelated to the experimental treatment was excluded from the final analysis to ensure data accuracy and reliability.

### 2.2. Construction of Full-Length Genome Plasmids of Recombinant Measles Virus

The full genome of the measles virus vaccine strain Schwarz was inserted into the pAC plasmid using a 5-fragment cloning and restriction enzyme ligation method to generate the full-length plasmid pSchwarz. The N, P, and L genes of the Schwarz strain were cloned separately into the pAC vector, resulting in the creation of helper plasmids pAC-N, pAC-P, and pAC-L. The F gene fragment of the MV-1 strain was amplified using the primers MV-F-F-XIN (5′-GAAAGACTCCATGGATCAAGCAATCAACCAGCCAGCAGCCAACGG-3′) and F-R-XIN (5′-GGATGATCTTGCACCCTAAGTTTTTATTAATGACCGATCGTATTCGGC-3′). The H gene fragment of the MV-1 strain was amplified with the primers H-F-XIN (5′-CTTAGGGTGCAAGATCATCCACAATGTCACCACAACGAGACCGG-3′) and H-R-XIN (5′-GTTGACAGATAGCGAGTCCATAACGGGGAACCACTTGGACCCTACG-3′). The F + H gene fragment of MV-1 was amplified using the primers MV-F-F-XIN and H-R-XIN. Through homologous recombination, the F, H, and F + H genes of MV-1 were, respectively inserted into the corresponding positions in pSchwarz to generate the recombinant full-length plasmids: pSchwarz/F(H1a), pSchwarz/H(H1a), and pSchwarz/FH(H1a).

### 2.3. Rescue of Recombinant Measles Virus

The full-length plasmid (4 μg) was co-transfected into BSR-T7 cells along with the helper plasmids pcDNA3.1-N (1.5 μg), pcDNA3.1-P (0.2 μg), and pcDNA3.1 L (1.0 μg) using Lipofectamine™ 2000 obtained from Invitrogen (Carlsbad, CA, USA). After six hours post-transfection, the cell culture medium was replaced. A total of 3 days later, the cells were passaged at a 1:3 ratio and co-cultured with an equal number of Vero-hSLAM cells until cytopathic effects were observed.

### 2.4. Growth Characteristics of Recombinant Measles Virus

Vero and CEF cells were seeded into 12-well plates at a density of 4 × 10^5^ cells per well and incubated overnight at 37 °C. When the cell confluence reached approximately 90%, the cells were washed thrice with PBS and then infected with the virus at an MOI of 0.01. After an hour of incubation at 37 °C, the inoculum was removed, and the cells were rinsed twice with PBS. Subsequently, the cell culture medium was replaced, and the cells were cultured at 37 °C in a 5% CO_2_ atmosphere. Samples were collected every 24 h for five consecutive days, and the virus samples were 10-fold serially diluted, and 100 μL of each dilution was added to 96-well plates pre-seeded with Vero-hSLAM cells (1.4 × 10^5^ cells/mL, 100 μL per well). Each dilution was tested in eight replicate wells. Following inoculation, the plates were incubated at 37 °C under a 5% CO_2_ atmosphere for 7 days. Cytopathic effects (CPE) were monitored and observed daily using an inverted optical microscope. The CCID_50_ (cell culture infectious dose 50%) was determined using the Reed–Muench method based on wells with observed CPE.

### 2.5. Scanning Electron Microscopy

A 500 mL virus suspension was centrifuged at 8000× *g* for 5 min to remove debris, after which the supernatant was collected. The virus supernatant was concentrated from 500 mL to 14 mL using the Pellicon XL Ultrafiltration Module, Biomax 100 kDa, 0.005 m^2^ (Merck, Rahway, NJ, USA, Catalog No.: PXB100C50). Next, the concentrated virus (13 mL), 30% sucrose solution (12 mL), and 60% sucrose solution (6 mL) were layered in sequence into a 31 mL centrifuge tube (Beckman, Brea, CA, USA, 355631) and centrifuged at 30,000× *g* for 14 h. The protein band, located at the 30–60% sucrose interface, was collected. A 20 μL aliquot of the sample was applied to a copper grid and left to adsorb naturally for 5–10 min, followed by air-drying. Subsequently, 20 μL of a 2% tungstic acid solution was added to the grid, allowed to stand for 3–5 min, and then air-dried. The prepared sample was examined and photographed using a JEM-1200EX transmission electron microscope (JEOL Ltd., Tokyo, Japan).

### 2.6. Immunization of Mice

Female BALB/c mice (Vital River Labs), aged 6–7 weeks and weighing 19–21 g, were divided into two groups (n = 10 per group). Each group was intraperitoneally immunized on days 0 and 14 with either the Schwarz strain or the MV-1 strain of measles virus at a dose of 1 × 10^5^ CCID_50_ per animal. After 14 days post-second immunization, serum samples were collected from all mice for the determination of neutralizing antibody titers against the Schwarz and MV-1 strains of measles virus.

Intraperitoneal immunization of six- to seven-week-old, 19–21 g female BALB/c mice with the recombinant measles virus rSchwarz, rSchwarz/F(H1a), rSchwarz/H(H1a), and rSchwarz/FH(H1a) at a dose of 1 × 10^5^ CCID_50_ per mouse on days 0 and 14; 10 mice were immunized per group. A total of 14 days after the second immunization, sera were collected to measure the neutralizing antibody titers against the Schwarz and MV-1 viruses.

Female BALB/c mice, aged 6–7 weeks and weighing 19–21 g, were intraperitoneally immunized with recombinant measles virus rSchwarz, rSchwarz/H (P397L + N405S), rSchwarz/H(F476L), and rSchwarz/H (P397L + N405S + F476L) at a dose of 5 × 10^4^ CCID50 per animal on days 0 and 14. Each group contained 5 mice. A total of 14 days after the second immunization, serum samples were collected to determine the neutralizing antibody titers against the Schwarz and MV-1 viruses. The experimental animals were euthanized using a CO_2_ euthanasia chamber.

Female BALB/c mice, aged 6–7 weeks and weighing 19–21 g, were intraperitoneally immunized with either the rSchwarz or rSchwarz/FH (H1a) viruses at high and low doses on days 0 and 14 (n = 10 per group). The high dose was set at 1 × 10^5^ CCID_50_ per animal, while the low dose was 1 × 10^3^ CCID_50_ per animal. A total of 14 days after the second immunization, serum samples were collected to determine the neutralizing antibody titers against the Schwarz, D8, MV-1, and rSchwarz/FH (B3) measles virus. The experimental animals were euthanized using a CO_2_ euthanasia chamber.

### 2.7. Immunization of Guinea Pigs

Female guinea pigs (Vital River Labs), weighing 250–300 g, were also intraperitoneally immunized with either the Schwarz or MV-1 strains of measles virus at a dose of 1 × 10^5^ CCID_50_ per animal on days 0 and 14. Each group contained five guinea pigs. A total of 14 days after the second immunization, serum samples were collected to determine the neutralizing antibody titers against the Schwarz and MV-1 viruses. The experimental animals were euthanized using a CO_2_ euthanasia chamber.

### 2.8. Challenge of IFNα/βR^−/−^ (A129) Mice

Five to six-week-old IFNα/βR^−/−^ (A129) mice were intracranially injected with 10^10^ vg (viral genomes) per animal of rAAV-hsyn-SLAM. After 60 days post-injection, the mice received an intracranial challenge with 30 μL of measles virus solution at a concentration of 3 × 10^4^ CCID_50_ per animal. All groups (n = 4 per group) were challenged once, and the day of the first challenge was defined as day 0 (D0). The animals in each experimental group were monitored for body weight after 14 days post-challenge and for mortality after 22 days post-challenge. The experimental animals were euthanized using a CO_2_ euthanasia chamber.

### 2.9. Immunization of Rhesus Macaques

Measles antibody-negative rhesus macaques (Suzhou Zhaoyan, Suzhou, China) were randomly divided into three groups based on body weight: a DMEM control group (n = 4), an S191 group (n = 10), and an rSchwarz/FH(H1a) group (n = 10). Each animal received a single bilateral intrathalamic injection of 1 mL of measles virus solution or DMEM culture medium, with a titer of log 1.78 × 10^4^ CCID_50_ per animal. Post-injection, animals were clinically observed and their body weights were measured for 31 days. On day 21 post-injection, serum samples were collected from each group to determine the neutralizing antibody titers against the Schwarz, MV-1, and D8 measles virus. The experiment was terminated on day 31 post-injection, at which point the animals were euthanized for histopathological examination of brain and spinal cord tissues.

The rhesus monkeys used in the experiment were fasted but not water-deprived on the day prior to surgery. On the day of the surgery, the animals were anesthetized via intravenous injection of propofol emulsion injection (Jiangsu Nhwa Pharmaceutical Co., Ltd., Xuzhou, China) at an induction dose of 5 mg/kg/min and a maintenance dose of 0.3–0.5 mg/kg/min. In accordance with the AVMA Guidelines for the Euthanasia of Animals: 2020 Edition, the experimental animals were euthanized at the conclusion of the experiment using a method that involved intramuscular injection of ketamine (10 mg/kg, 50 mg/mL) combined with xylazine hydrochloride (2 mg/kg, 10 mg/mL) for anesthesia, followed by euthanasia via femoral artery exsanguination.

### 2.10. Animal Ethics Statement

The evaluation experiments based on small animal models such as mice in this study were approved by the Experimental Animal Welfare Ethics Committee of the Jiangsu Provincial Center for Disease Control and Prevention, with approval number NO. JSJK/JL-161.

The evaluation experiment based on the rhesus monkey model in this study was conducted by the JOINN (Suzhou) New Drug Research Center Co., Ltd. (Suzhou, China), The care and use of laboratory animals were conducted following the Guide for the Care and Use of Laboratory Animals, 8th Edition (Institute of Laboratory Animal Resources, Commission on Life Sciences, National Research Council; National Academy Press, Washington, DC, USA, 2011) and the Animal Welfare Regulations by the United States Department of Agriculture (USDA) as stipulated by Public Law 99–198. The study was performed at the Zophan (Suzhou) New Drug Research Center Co., Ltd. (Suzhou, China), an institution accredited by the Association for Assessment and Accreditation of Laboratory Animal Care (AAALAC International). All activities involving animal use and welfare were approved by the Institutional Animal Care and Use Committee (IACUC) prior to operation. The IACUC approval number is S-ACU23-0719.

All procedures, including vaccination, sample collection, and challenge experiments, were performed under appropriate anesthesia. Every effort was made to minimize animal suffering and to use the minimum number of animals required to achieve statistically valid results.

### 2.11. Micro-Neutralization Assay

Serum samples were heat-inactivated for 30 min at 56 °C. An amount of 50 μL of DMEM (Gibco, Grand Island, NY, USA) were dispensed into each well of a 96-well plate, followed by the addition of 50 μL of test sera serially diluted 2-fold, with four replicate wells established for each dilution. Subsequently, 50 μL of a measles virus solution at a concentration of 2000 CCID_50_/mL were added to each well and thoroughly mixed. After incubating the plates at 37 °C for 1 h, 100 μL of a Vero-hSLAM cell suspension at a concentration of 2.0 × 10^5^ cells/mL were added to each well. The plates were then incubated at 37 °C in a 5% CO_2_ atmosphere for 7 days, and the NT_50_ (50% neutralization titer) was calculated using the Reed–Muench method.

### 2.12. Viral Genome Sequencing and Bioinformatic Analysis

Viral samples of rSchwarz/FH(H1a) at passage P0, P7, P9, P12, P17, and P22 in chicken embryo fibroblast (CEF) cells were submitted to Shanghai Sequanta Technologies Co., Ltd. (Shanghai, China) for next-generation sequencing (NGS)-based analysis to determine the viral genome sequences. Briefly, total nucleic acids were extracted from each sample, followed by reverse transcription to synthesize complementary DNA (cDNA). The cDNA was then subjected to poly-A tailing, adapter ligation, and PCR amplification to generate RNA sequencing libraries. The sequencing was performed using an Illumina (San Diego, CA, USA) Novaseq 6000 PE 150 platform. The resulting sequence data were subjected to bioinformatics assembly processes to re-construct the complete full-length genomic sequences of the virus.

### 2.13. Statistical Analysis

All statistical analyses and graphical representations were performed using GraphPad Prism software (version 8.0). Data for viral growth kinetics, longitudinal changes in body weight and temperature of mice and nonhuman primates (NHPs), and the durability of immunogenicity induced by candidate vaccine strains were expressed as mean ± standard error of the mean (SEM). Neutralizing antibody titers from mice, guinea pigs, NHPs, and human serum samples were presented as mean with 95% confidence intervals (95% CI). Kaplan–Meier survival curves were analyzed by the log rank test to make statistical comparisons between groups for experiments and to determine the percentage of survival. For all tests, the following notations were used to indicate significant differences between groups: ns (Not Significant): *p* ≥ 0.05, *: 0.01 ≤ *p* < 0.05; **: 0.001 ≤ *p* < 0.01; ***: *p* < 0.001.

## 3. Results

### 3.1. Differential Immunogenicity Between Measles Virus Genotypes A and H1a: Genotype-Specific Neutralization Potency of Post-Immunization Sera

To investigate the immune cross-protection between H1a and A genotypes of measles virus, BALB/c mice and guinea pigs were immunized with 1 × 10^5^ CCID_50_ of MV-1 (H1a genotype) and rSchwarz (A genotype), respectively. Serum samples were collected 14 days after the second immunization, and specific neutralizing antibody titers against both genotypes were determined (Figure 1A,B). Significantly higher neutralizing antibody titers were observed against the homologous genotype compared to the heterologous genotype after immunization with either MV-1 or rSchwarz. Additionally, a total of 49 serum samples were obtained from 18-month-old infants in Changzhou, Jiangsu Province, China, one month following immunization with the measles-mumps-rubella (MMR) vaccine developed by the Beijing Institute of Life Sciences. The samples were used to determine neutralizing antibody titers specific to different measles virus genotypes. This study was approved by the Ethics Committee of Jiangsu Provincial Center for Disease Prevention and Control (No. JSJK2020-B005-01). As shown in Figure 1C, the neutralizing antibody titers in post-MMR immunization sera against the A genotype virus were notably higher than those against the H1a genotype virus. Collectively, these results indicate a remarkable finding of differential immunogenicity between Measles Virus Genotypes A and H1a: genotype-specific neutralization potency of post-immunization sera.

### 3.2. rSchwarz/FH(H1a) Exhibits Improved Immunogenicity, with the H Protein Residue 476 Playing a Key Role in Cross-Protection Between Genotypes H1a and A Measles Virus

Using the Schwarz strain as the parental virus, a series of recombinant chimeric measles virus, by replacing the gene sequences encoding the viral surface proteins (F and H) with those from the H1a genotype virus (Figure 2A), were constructed and evaluated. Initially, as shown in Figure 2B, no significant difference in particle size between the recombinant virus rSchwarz/FH(H1a) and the parental virus rSchwarz was observed. Furthermore, the cytopathic effects induced by the recombinant viruses in CEF and Vero cells were comparable to those of the parental virus (Figure 2C). Multi-step growth curves demonstrated that the replacement of the F and/or H protein genes did not significantly alter the replication characteristics of the recombinant viruses in Vero cells and CEF (Figure 2D,E). Additionally, immunogenicity evaluations in mice showed that the titers of neutralizing antibodies induced by the three recombinant viruses (rSchwarz/F(H1a), rSchwarz/H(H1a), rSchwarz/FH(H1a)) against both H1a and A genotype viruses were higher than those induced by the parental rSchwarz strain (Figure 2F). This indicates that the H protein plays a pivotal role in the different cross-protection observed between H1a and A genotypes of measles virus.

We conducted a comparative analysis of the H protein amino acid sequences between H1a and A genotype of measles virus, followed by antigenic epitope predictions. Utilizing reverse genetics, amino acids at positions 397, 405, and 476 of the Schwarz H protein were mutated to match those of the wild-type H1a measles virus H protein, and recombinant virus were rescued and evaluated (Figure 3A). Multi-step growth kinetics in Vero cells revealed no significant differences in the replication capacity of the recombinant virus following mutations at amino acid positions 397, 405, and 476 (Figure 3B). Strains rSchwarz/H (P397L + N405S) and rSchwarz/H (P397L + N405S + F476L) induced significantly higher neutralizing antibodies against genotype A than against H1a measles virus in mice, similar to strain rSchwarz. Strain rSchwarzH/(F476L), however, showed no significant difference in neutralizing activity against the two genotypes (Figure 3C). This indicates that the H protein residue 476 plays a key role in cross-protection between genotypes H1a and A measles virus.

### 3.3. rSchwarz/FH(H1a) Induces High Titers of Neutralizing Antibodies Against Multiple Epidemic Genotype Strains of Measles Virus Compared to rSchwarz

To assess the immunogenicity of rSchwarz/FH(H1a) as a vaccine candidate against B3 and D8 genotype measles virus, BALB/c mice were immunized with rSchwarz/FH(H1a) at doses of 10^3^ CCID_50_ and 10^5^ CCID_50_/mL, respectively (Figure 4A). After 14 days post-immunization (dpi), serum samples were collected and titers of specific neutralizing antibodies against H1a, B3, D8, and A genotype measles virus were tested. As shown in Figure 4B, the titers of neutralizing antibodies induced by rSchwarz/FH(H1a) against all four genotypes (H1a, B3, D8, and A) were significantly higher than those of the parental rSchwarz strain, demonstrating the stronger immunogenicity of rSchwarz/FH(H1a) compared to others.

### 3.4. rSchwarz/FH(H1a) Exhibits Good Genetic Stability and Immune Persistence

Genetic stability is a critical parameter for evaluating vaccine candidates. In this study, we assessed the genetic stability of the rSchwarz/FH(H1a) vaccine candidate by performing continuous passages, followed by whole-genome sequencing and sequence alignment analysis across multiple passages (Figure 5A). Genetic stability tests across 22 passages confirmed consistent immunogenicity and replication profiles for rSchwarz/FH(H1a), ensuring its reliability as a vaccine candidate. Compared to P0, a single amino acid mutation was detected in L proteins of measles virus following seven consecutive serial passages, yet no further base mutations were observable from P7 to P22. To investigate the influence of these mutations on the viral growth properties, multi-step growth curve analyses of the virus were conducted in CEF cells (Figure 5B). The results indicated no significant changes in the replication characteristics of the long-term passaged rSchwarz/FH(H1a). Furthermore, BALB/c mice were immunized with 1 × 10^5^ CCID_50_ of rMV/FH(H1a) derived from passages P1, P12, and P22. Serum samples were collected from the mice 14 dpi, and the titers of specific neutralizing antibodies against H1a, D8, and A genotype measles virus were assayed. As depicted in Figure 5C, the immunogenicity of rMV/FH(H1a) remained consistent across passages P1, P12, and P22, showing no significant variations. We also conducted an experiment to assess the immune durability of rSchwarz/FH (H1a) in BALB/c mice. Six-week-old female BALB/c mice were immunized intraperitoneally with 1 × 10^5^ CCID_50_ of rSchwarz/FH (H1a). Peripheral blood was collected at various time points post-immunization, and serum was tested for measles neutralizing antibodies. As shown in Appendix A, the titers of anti-A genotype and anti-H1a genotype neutralizing antibodies in mouse serum remained at high levels six months post-immunization with no significant decline observed.

### 3.5. rSchwarz/FH(H1a) Exhibits Valid Safety in Mice and Rhesus Macaques

To further evaluate the safety of rSchwarz/FH(H1a) in vivo, we conducted challenge experiments on mice IFNα/βR^−/−^ (A129) (Figure 6A). The findings (Figure 6B) demonstrated that the weight loss ratio of mice challenged with rSchwarz/FH(H1a) was comparable to that of mice challenged with the parental virus, rSchwarz, and markedly lower than that of mice challenged with the wild-type MV-1 strain (H1a genotype). However, MV-1 induced 75% mortality and rSchwarz resulted in 50% mortality, whereas all mice challenged with rSchwarz/FH(H1a) survived (Figure 6C).

Moreover, measles antibody-negative rhesus monkeys were intracranially inoculated with rSchwarz/FH(H1a) and S191 to evaluate viral neurovirulence (Figure 7A). The immunogenicity of rSchwarz/FH(H1a) was evaluated in rhesus monkeys by intracranial injection of 1.78 × 10^4^ CCID50/mL of rSchwarz/FH(H1a) and S191 (A genotype national standard in China), with DMEM injection serving as a control. Serum samples were collected at 21 dpi, and titers of specific neutralizing antibodies against H1a, D8, and A genotype measles virus were tested. As shown in Figure 7B,C, both groups of monkeys infected with rSchwarz/FH(H1a) and S191 had 100% seropositivity rates, but the rSchwarz/FH(H1a) infection group exhibited significantly higher neutralizing antibody titers against all three genotype viruses compared to the S191 infection group. Compared to the control group of monkeys, there were no significant differences in animal weights between all groups. Additionally, there were no remarkable differences in body temperatures between infection groups (Figure 7D,E). Furthermore, histopathological examination further confirmed that rSchwarz/FH(H1a) infection did not result in virus-related neural damage in the brains of monkeys (Figure 7F).

## 4. Discussion

Over the past two decades, surveillance data from the World Health Organization (WHO) has indicated an increasing prevalence of the H1a, B3, and D8 measles virus genotypes in circulation [26]. Studies conducted in China have demonstrated that the H1a genotype is the predominant strain circulating in the country [27]. The phenomenon of different cross-protection among different measles virus genotypes has garnered considerable attention. Previous research has shown different cross-protection between the A genotype and the D4, D8, and B3 genotypes [24,25]. In this study, through immunization experiments in mice and guinea pigs, we further confirmed the existence of different cross-protection between the currently circulating H1a, B3, and D8 measles virus strains and the A genotype measles vaccine strain. A study of measles cases in Tianjin, China, reported that more than 25% of cases occurred in individuals who had received at least one dose of the measles vaccine prior to infection [28]. Several factors could contribute to high breakthrough infection rates, including issues related to vaccine storage, transportation, administration techniques, waning immunity in the vaccinated population, and insufficient immunization coverage. However, another critical aspect to consider is the antigenic drift between the circulating strains and the vaccine strains used. The divergence in antigenicity between prevalent virus strains and the vaccine strains can significantly impact vaccine efficacy and may explain some of the observed breakthrough infections [29,30]. Therefore, the development of attenuated vaccines targeting the newly emerging measles virus genotypes is of utmost importance. In this investigation, we systematically evaluated the cross-protection provided by the vaccine candidate strain rSchwarz/FH(H1a), which exhibited effective cross-protection against H1a, B3, and D8 genotype viruses. Moreover, its immune efficacy surpassed that of the parental vaccine strain rSchwarz, indicating that this vaccine candidate possesses notable immunogenicity against these genotypes.

The primary immunogenic protein of measles virus is the H protein, and existing research has established that multiple neutralizing antibody epitopes are located on this protein, potentially contributing to the observed different cross-protection among different genotype viruses [31,32,33,34,35]. Prioritizing safety for the vaccine strain, our initial objective in this study was to utilize the Schwarz strain as the backbone and substitute key specific neutralizing epitopes in the H protein with those of the H1a genotype measles virus. Through antigenic epitope prediction, we successfully constructed three recombinant measles virus: rSchwarz/H (P397L + N405S), rSchwarz/(F476L), and rSchwarz/H (P397L + N405S + F476L). During the study, we identified that the 476 epitope plays a critical role in cross-protection between genotype A and H1a measles virus. However, the immunogenicity results in mice also demonstrated that the neutralizing antibody titers induced by these three recombinant viruses against the H1a genotype were significantly lower compared to those induced by virus rSchwarz/H(H1a). Given the many neutralizing epitopes on the measles virus H protein, more research is needed to fine-tune them and enhance the Schwarz vaccine strain’s immunity to match that of the H1a genotype virus.

In the course of measles vaccine development, inactivated measles vaccines were temporarily employed; nevertheless, their effectiveness was constrained by factors including diminished immunogenicity, transient immunity, and notable side effects [36]. The development of attenuated measles vaccines, which effectively stimulate humoral, cellular, and mucosal immune responses, has played a pivotal role in curbing measles outbreaks. Over several decades of utilization, these vaccines have earned widespread acclaim as one of the most efficacious attenuated vaccines against respiratory viruses and have established themselves as the cornerstone of measles vaccine research and development [37]. Currently, all measles vaccine products available worldwide are based on attenuated live virus technology. Therefore, the development of novel measles vaccines should continue to prioritize advancements in attenuated live vaccine technology.

Previous research has demonstrated that the P and M genes are among the primary determinants of measles virus virulence [38,39,40,41,42]. Although certain studies have indicated that the F and H genes also display characteristics of virulence, these traits are specific to particular strains [43,44,45]. Whether rSchwarz/FH (H1a) can inherit the safety characteristics of its parental virus is a matter of utmost concern to us. To better predict the neurovirulence risk of the recombinant virus, IFNα/βR^−/−^ mice were infected with an AVV virus expressing human SLAM via intracranial inoculation and subsequently were challenged intracranially with equal doses of the recombinant virus. The choice of animal model was based on commonly used transgenic mouse models for studying measles virus pathogenicity, including CD46-overexpressing transgenic mice [46], SLAM-overexpressing transgenic mice [47], and IFNα/βR^−/−^ mice [48]. When IFNα/βR^−/−^ mice immunized with Schwarz vaccine strain were challenged with measles virus, viral replication could be detected from samples at the injection site. Additionally, attenuated strain infection induced cells to produce more interferon, which significantly inhibits the replication of wild-type strains [49]. Given that SLAM serves as a common receptor for both wild-type and vaccine strains of measles virus, we selected IFNα/βR^−/−^ mice expressing the SLAM receptor to compare the virulence of measles virus strains. However, constructing stable SLAM-overexpressing IFNα/βR^−/−^ mice is time-consuming and costly. So, we consideed the high and stable expression of exogenous proteins by AAV vectors by injecting rAAV-hsyn-SLAM virus intra-cranially into IFNα/βR^−/−^ mice to establish a model capable of stably expressing the SLAM receptor in the brain. All mice in the rSchwarz/FH (H1a) group survived, whereas 50% of mice in the parental Schwarz virus group succumbed to the infection. For added verification, we opted to utilize a rhesus monkey model for thalamic inoculation to evaluate the neurovirulence of the recombinant virus. The results revealed that monkeys in the rSchwarz/FH (H1a) challenge group, despite eliciting higher levels of genotype H1a-specific measles virus neutralizing antibodies, exhibited no notable differences in body temperature fluctuations, weight gain, or behavioral patterns when compared to monkeys immunized with the S191 strain (a licensed measles vaccine in China). Based on the results of these two animal evaluation experiments, it can be inferred that rSchwarz/FH (H1a) is a safe candidate vaccine strain for measles. Regarding the lower mortality rate observed in IFNα/βR^−/−^ mice infected with rSchwarz/FH (H1a) compared to the parental virus, we attribute this to two primary factors: the sample size utilized in the present study is relatively limited (n = 4) and the utilization of IFNα/βR^−/−^ mice model expressing human SLAM in the brain are highly sensitive for measles virus infection. In the future, we plan to consider further refining this infection model to better suit the needs of our subsequent research endeavors.

In summary, we developed a recombinant measles virus strain that provides immune protection against the H1a genotype as well as cross-protection against the B3 and D8 genotypes. Additionally, evaluation of virus stability and safety indicated that this recombinant measles virus strain holds promise as a potential vaccine candidate, potentially serving as a broad-spectrum vaccine to limit the infection and spread of the currently predominant measles virus.

## 5. Conclusions

A novel recombinant measles vaccine, rSchwarz/FH(H1a), derived from the Schwarz strain with immunogenic genes from the H1a genotype, demonstrates enhanced immunogenicity, genetic stability, and safety, providing cross-protection against multiple epidemic genotypes (H1a, B3, D8) and showing potential as a broad-spectrum vaccine candidate.

## Figures and Tables

**Figure 1 vaccines-13-00571-f001:**
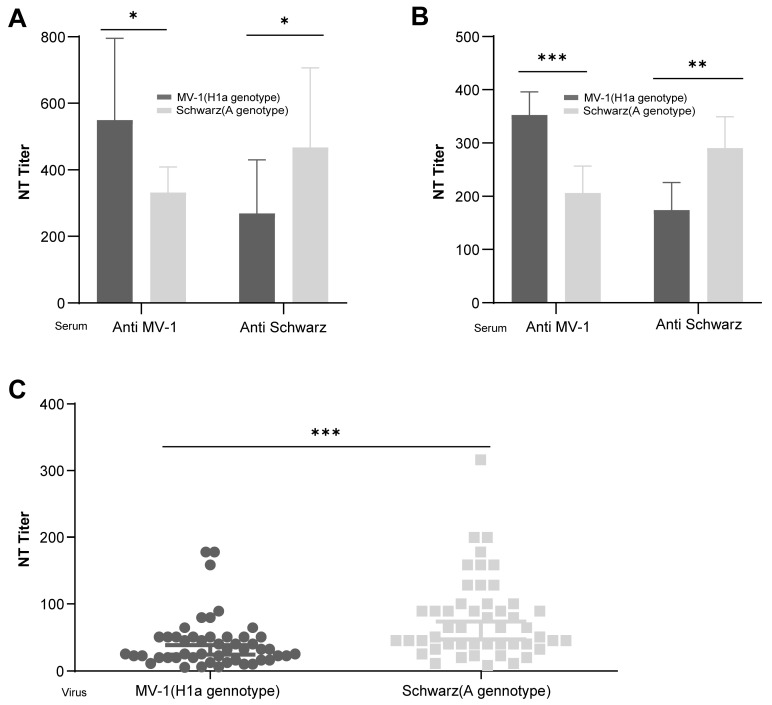
Cross-neutralization analysis of measles virus strains of H1a and A genotype in vivo. Female BALB/c mice as well as female guinea pigs were intraperitoneally injected with Schwarz and MV-1 virus strains at a dose of 1 × 10^5^ CCID_50_, and second immunization were carried out after 14 days. Each group contained 10 mice or 5 guinea pigs. A total of 14 days after the second immunization, serum samples were collected to determine the neutralizing antibody titers against the Schwarz and MV-1 viruses. The graphs show the geometric mean antibody titers for each group of guinea pigs (**A**) or mice (**B**). All animals underwent blood collection and neutralization titer testing prior to immunization. Following a 1:1 (original) dilution of serum from all experimental animals, no viral inhibitory activity was observed in the microneutralization assay, as indicated by 100% cytopathic effect (CPE) while all samples exhibited a titer of <2. Titers of antibody against the Schwarz and MV-1 viruses in sera samples from 18-month-old infants who have received the MMR vaccine were determined by micro-neutralization antibody test (**C**). All data are expressed in mean with 95% confidence intervals (95% CI). *: 0.01 ≤ *p* < 0.05; **: 0.001 ≤ *p* < 0.01; ***: *p* < 0.001.

**Figure 2 vaccines-13-00571-f002:**
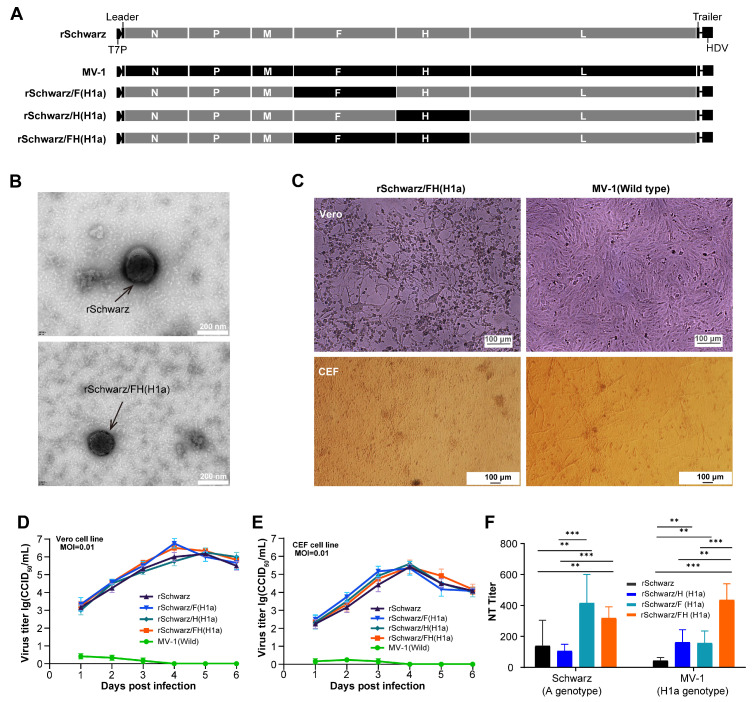
Characterization of Measles Virus in which F and H structures were modified. (**A**) Schematic diagrams show the gene structures of the recombinant measles virus. Gene F or/and H on Schwarz strain were replaced with segments of MV-1 virus to produce recombinant rSchwarz, rSchwarz/F(H1a), rSchwarz/H(H1a), and rSchwarz/FH(H1a). (**B**) The rSchwarz and rSchwarz/FH(H1a) were observed under electron microscopy. (**C**) Observation of cytopathic effect (CPE) on measles virus recombinants were performed by infection of Vero and CEF cell lines with gene-modified viruses at a multiplicity of infection (MOI) of 0.01, and cells were imaged under a 10X wide field microscope after four days post-infection. Growth curves of the recombinant viruses on Vero (**D**) and CEF (**E**) cells were analyzed by infecting the cells with viruses at 0.01 MOI, and viral titers were determined by using Vero-hSLAM cells incubated with the collected supernatants harvested every 24 h until up to 144 h of infection. (**F**) In vivo, each of the 10 female BALB/c mice were intraperitoneally injected with rSchwarz, rSchwarz/F(H1a), rSchwarz/H(H1a), and rSchwarz/FH(H1a) at a dose of 1 × 10^5^ CCID_50_, followed by a second immunization after 14 days. After two weeks of second immunization, mouse sera were collected to measure the neutralizing antibody titers against the Schwarz and MV-1 viruses, respectively. The neutralizing antibody titers were evaluated in serum samples collected from all animals before immunization, and all samples exhibited a titer of <2. Data for viral growth kinetics were expressed as mean ± standard error of the mean (SEM), while neutralizing antibody titers are expressed in mean with 95% confidence intervals (95% CI). **: 0.001 ≤ *p* < 0.01; ***: *p* < 0.001.

**Figure 3 vaccines-13-00571-f003:**
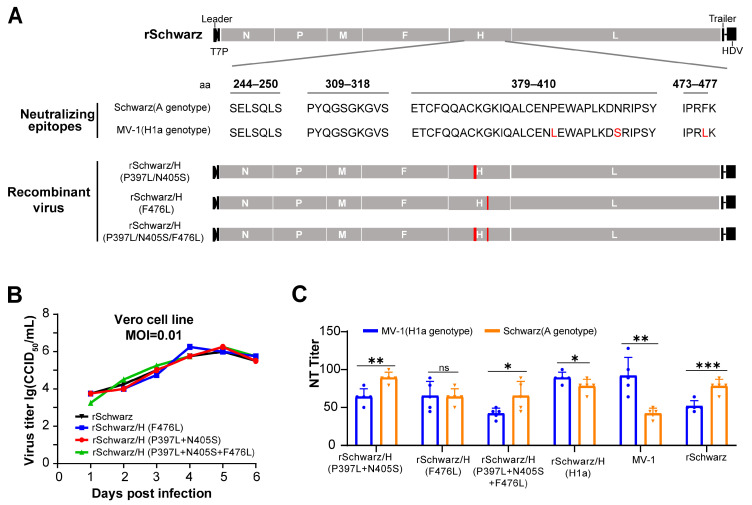
The amino acid at position 476 of the H protein is a critical determinant for different cross-protection between measles virus strains of genotypes H1a and A. (**A**) Schematic representation of the construction of recombinant measles virus rSchwarz/H (P397L + N405S), rSchwarzH/(F476L), and rSchwarz/H (P397L + N405S + F476L). This includes detailed amino acid sequences of key antigenic epitopes from both A and H1a genotype measles virus; mutations associated with the MV-1 strain are highlighted in red, while proteins related to the Schwarz strain are indicated in gray. (**B**) Multistep growth curves of the recombinant viruses in Vero cells. The recombinant viruses were used to infect Vero cells at an MOI of 0.01, and viral supernatants were harvested every 24 h for up to 144 h. Virus titers at each time point were measured in Vero-hSLAM cells. (**C**) Female BALB/c mice, aged 6–7 weeks and weighing 19–21 g, were intraperitoneally immunized with recombinant measles virus rSchwarz, rSchwarz/H (P397L + N405S), rSchwarzH/(F476L), and rSchwarz/H (P397L + N405S + F476L) at a dose of 5 × 10^4^ CCID_50_ per animal on days 0 and 14. Each group contained five mice. A total of 14 days after the second immunization, serum samples were collected to determine the neutralizing antibody titers against the Schwarz and MV-1 viruses. Data for viral growth kinetics were expressed as mean ± standard error of the mean (SEM), neutralizing antibody titers are expressed in mean with 95% confidence intervals (95% CI). ns (Not Significant): *p* ≥ 0.05, *: 0.01 ≤ *p* < 0.05; **: 0.001 ≤ *p* < 0.01; ***: *p* < 0.001. The neutralizing antibody titers were assessed in sera collected from all animals prior to immunization, with all samples showing a titer of <2.

**Figure 4 vaccines-13-00571-f004:**
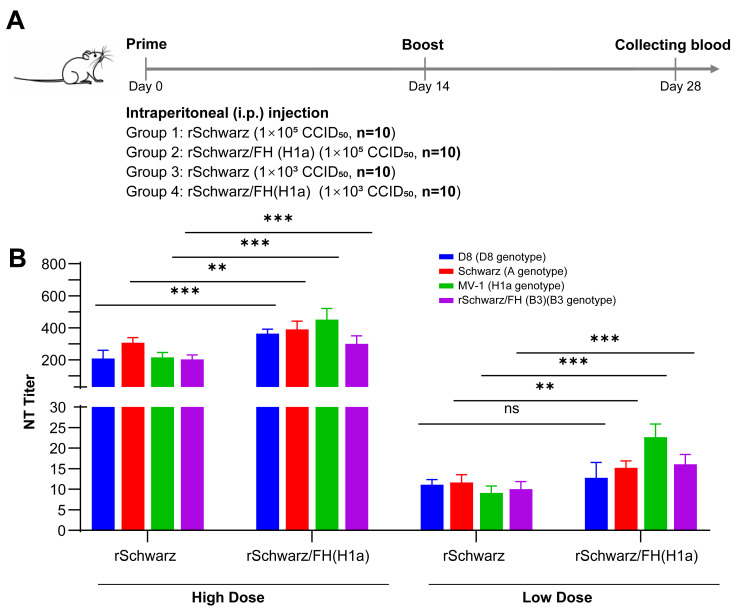
(**A**) Evaluation of the immunogenicity of recombinant measles virus against multiple genotypes. Each of the 10 female BALB/c mice, aged at 6–7 weeks and weighed between 19 and 21 g, were intraperitoneally immunized with either the rSchwarz or rSchwarz/FH (H1a) viruses at high (1 × 10^5^ CCID_50_) and low (1 × 10^3^ CCID_50_) doses on day 0 and a second immunization on day 14. A total of 14 days after the second immunization, serum samples were collected from the mice. (**B**) Neutralizing antibody titers of serum samples collected from immunized mice were determined by micro-neutralization antibody test. The graphs show the geometric mean antibody titers for each group of mice. The neutralizing antibody titers were assessed in sera collected from all animals prior to immunization, with all samples showing a titer of <2. All data are expressed in mean with 95% confidence intervals (95% CI). ns (Not Significant): *p* ≥ 0.05, **: 0.001 ≤ *p* < 0.01; ***: *p* < 0.001.

**Figure 5 vaccines-13-00571-f005:**
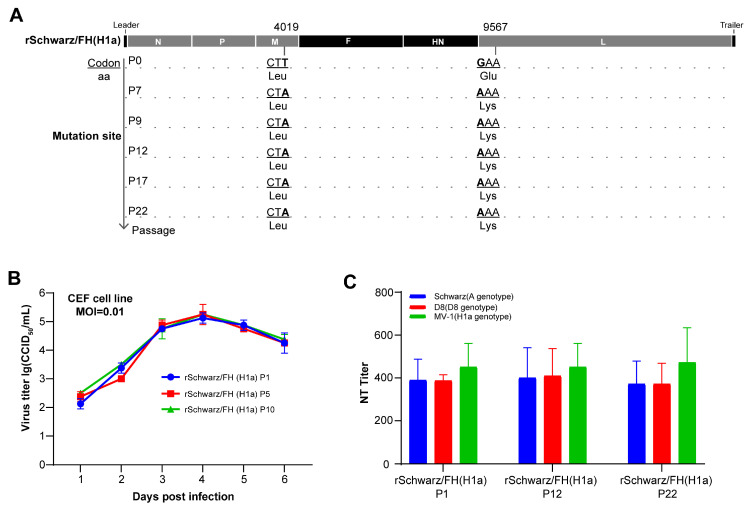
Genetic stability of rSchwarz/FH (H1a) as a vaccine candidate. (**A**) The rSchwarz/FH (H1a) virus was continuously passaged for 10 passages in primary chicken embryo fibroblast (CEF) cells. Viral samples at passages P0, P7, P9, P12, P17, and P22, were collected, followed by full-genome sequencing using Sanger sequencing. Sequence alignment was performed to analyze positions and details of nucleotide mutations. (**B**) Growth characteristics of rSchwarz/FH (H1a) virus passaged in CEF cells were confirmed by infecting the CEF cells at 0.01 MOI followed by determination of virus titers by using Vero-hSLAM cells. (**C**) Each group of five female mice were intraperitoneally immunized with rSchwarz/FH (H1a) viruses from passages P1, P12, and P22 at a dose of 1 × 10^5^ CCID_50_ on day 0 and 14. A total of 14 days after the second immunization, serum samples were collected from the mice, and titers of neutralizing antibody titers against the Schwarz, D8 and MV-1 measles virus were determined by micro-neutralization antibody test. The neutralizing antibody titers were assessed in sera collected from all animals prior to immunization, with all samples showing a titer of <2. Data for viral growth kinetics were expressed as mean ± standard error of the mean (SEM), while neutralizing antibody titers are expressed in mean with 95% confidence intervals (95% CI).

**Figure 6 vaccines-13-00571-f006:**
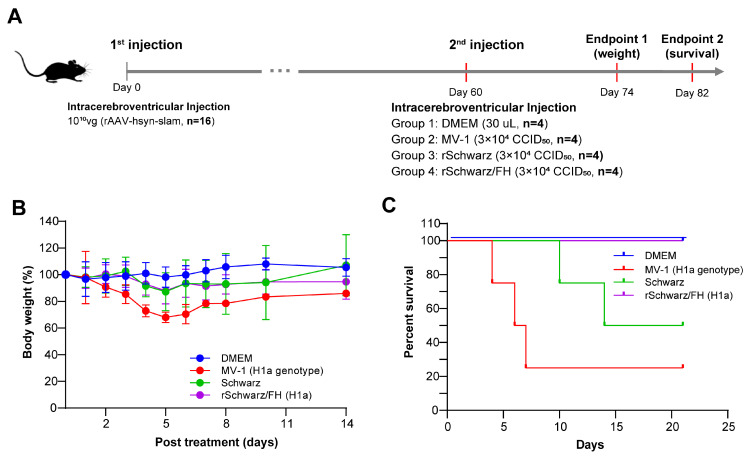
Safety of rSchwarz/FH (H1a) as a vaccine candidate in Mice. (**A**) IFNα/βR^−/−^ (A129) mice (5–6 weeks old) were intracranially administered 10^10^ vg rAAV-hsyn-SLAM, followed 60 days later by a single intracranial challenge with 30 μL of measles virus (3 × 10^4^ CCID_50_). Each group contained four animals. Body weight was monitored for 14 days post-challenge (D0), and mortality was observed for 22 days. (**B**) Statistical analysis of mouse body weight changes 14 days after challenge in the DMEM, MV-1, rSchwarz, and rSchwarz/FH(H1a) groups. (**C**) Mortality statistics 22 days after challenge in the DMEM, MV-1, rSchwarz, and rSchwarz/FH(H1a) groups.

**Figure 7 vaccines-13-00571-f007:**
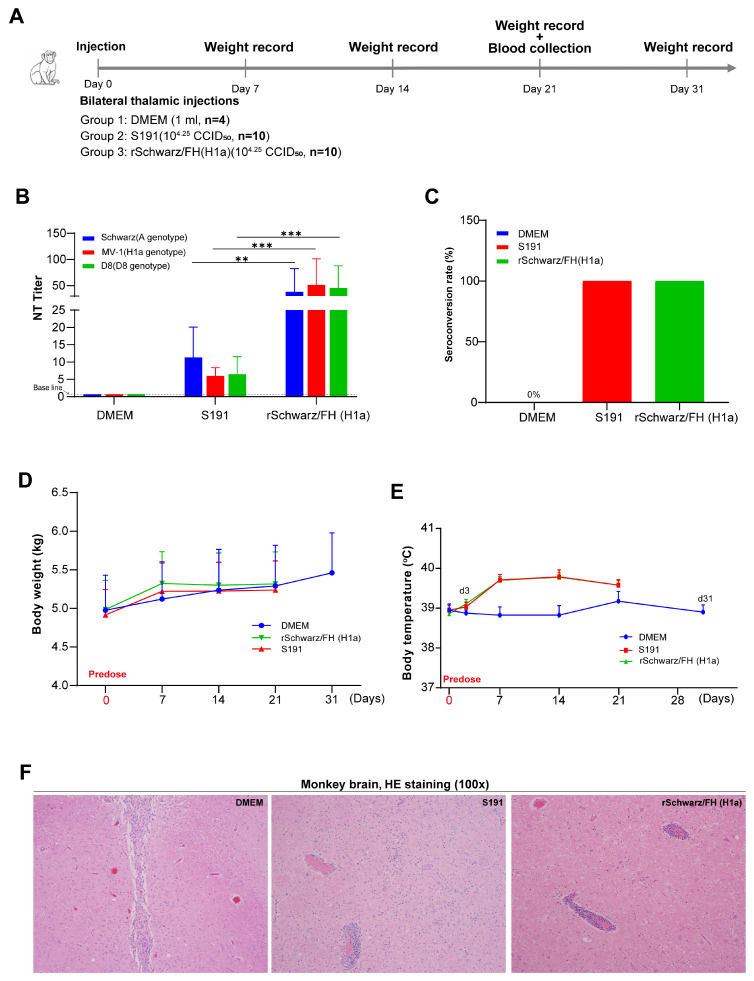
Safety of rSchwarz/FH (H1a) as a vaccine candidate in Rhesus Macaques. (**A**) Measles antibody-negative rhesus macaques were divided into three groups by body weight: DMEM control (n = 4), S191 (n = 10), and rSchwarz/FH(H1a) (n = 10). Each received a single bilateral intrathalamic injection of 1 mL solution (10^4.25^ CCID_50_ measles virus or DMEM). Clinical signs and body weights were monitored for 31 days. Serum samples for neutralizing antibody titers against Schwarz, MV-1, and D8 strains were collected on day 21. Animals were euthanized on day 31 for histopathological examination of brain and spinal cord. (**B**) Neutralizing antibody titers against the Schwarz, MV-1, and D8 measles virus were measured by micro-neutralization antibody test in serum samples collected from the DMEM, S191, and rSchwarz/FH(H1a) groups on day 21 post-injection. The graphs show the geometric mean antibody titers for each group of rhesus macaques. The neutralizing antibody titers were assessed in sera collected from all rhesus macaques prior to immunization, with all samples showing a titer of <2. And the neutralizing titers of the immune sera from the DMEM group were also all <2. Data for longitudinal changes in body weight and temperature of nonhuman primates (NHPs) were expressed as mean ± standard error of the mean (SEM). Neutralizing antibody titers were presented as mean with 95% confidence intervals (95% CI). **: 0.001 ≤ *p* < 0.01; ***: *p* < 0.001. (**C**) Statistical analysis of seroconversion rates, a neutralizing titer of >2 was considered positive. (**D**) Post-injection body weight changes in DMEM (monitored until day 31), S191, and rSchwarz/FH(H1a) (both monitored until day 21) groups compared. (**E**) Post-injection body temperature changes in DMEM (monitored until day 31), S191, and rSchwarz/FH(H1a) (both monitored until day 21) groups compared. (**F**) H&E-stained histological sections of brain and spinal cord tissues from monkeys injected with DMEM, S191, or rSchwarz/FH(H1a), examined at 100× magnification.

## Data Availability

The data presented in this study are available in this article and Appendix A.

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
