# Peer review of "Construction and Preclinical Evaluation of a Recombinant Attenuated Measles Vaccine Candidate of the H1a Genotype"

_vaccines, 2025, doi:10.3390/vaccines13060571_

Round 1
Reviewer 1 Report
Comments and Suggestions for Authors
The manuscript describes a recombinant measles virus developed using the Schwarz vaccine strain as a backbone, with the F and H genes replaced by those from a genotype H1a strain—a genotype that predominated in China prior to 2020. The authors demonstrate that the recombinant virus, designated Schwarz/FH(H1a), exhibits enhanced immunogenicity, genetic stability, and safety. It provides cross-protection against dominant epidemic genotypes (H1a, B3, D8) and shows promise as a potential improved vaccine candidate.
The manuscript is generally well written; however, several concerns remain regarding the clarity and completeness of the data presented:
- Organization of the Content: The manuscript’s readability and flow could be improved by reorganizing certain sections. For example, portions of Sections 3.1 and 3.3 contain background information that may be more appropriately placed in the Introduction, where they can provide context for the experimental rationale. Additionally, Supplementary Figure S2 contains data that are central to the findings discussed in Section 3.4 and should be included in the main text to enhance clarity.
- Data Interpretation: The authors should clearly define statistical significance while accounting for the inherent inter-assay variability of neutralization assays.
- Lack of Negative Controls and Baseline Data: The absence of pre-immunization or negative control data limits the ability to assess the true magnitude of the immune response and to make meaningful comparisons of cross-neutralizing titers across different strains.
- Comparability to Protective Thresholds: Although the protective threshold for measles immunity is defined as 120 mIU/mL of neutralizing antibodies, the manuscript does not clearly demonstrate whether the titers elicited by the recombinant virus meet or exceed this level in the various animal models.
- Lack of Challenge Studies: While the authors propose that the recombinant virus may be superior to existing vaccines, no challenge studies were conducted to confirm its protective efficacy in immunized animals following exposure to wild-type measles virus.
The following section provides detailed comments
A. General
- Please spell out abbreviations upon first use. For example, spell out CCID₅₀ as cell culture infectious dose 50% when it first appears in the manuscript.
- Please ensure consistent formatting throughout the manuscript. For instance, while most dose values are presented in standard notation, scientific notation is used in Line 187. Additionally, terminology should be uniform across figures. For example, use a single defined term such as NT titer consistently on the Y-axis rather than alternating between NT titer and NT₅₀.
B. Materials and Methods
1. 2.1. Cells and Viruses
1) Vero/hSLAM cells may require a license in China due to regulations involving dual-use biological agents and the Convention on International Trade in Endangered Species. Please confirm whether appropriate permissions were obtained.
2) Please provide complete information for all virus strains used (D8, B3, MV-1, Schwarz, S191), including strain names, GenBank accession number (if available), preparation methods, and passage numbers.
2. 2.5. Growth Characteristics of Recombinant Measles Virus
1) Consider merging Sections 2.5 and 2.7 (Virus Titer Determination) to streamline the methodology.
2) Clarify whether the temperature difference between virus adsorption (37 °C; Line 126) and subsequent incubation (35 °C; Line 128) is intentional.
3) Line 129: Expand on how virus titers were determined. Indicate that Vero-hSLAM cells were used (as per Line 324), and specify how cytopathic effects were assessed (e.g., crystal violet staining, neutral red, or light microscopy).
3. 2.6. Scanning Electron Microscopy
1) Line 133: Please describe the method used to concentrate supernatant from 500 mL to 14 mL.
4. 2.13. Micro-neutralization assay
1) Spell out ED50 on first mention and note that neutralizing antibody titers are typically reported as NT50 or nAb50 to denote 50% neutralization.
C. Results
- 3.1. The measles virus strains circulating in China are predominantly of the H1a genotype.
1) This section provides useful background but is not central to the main findings. Consider moving it to the Introduction.
2) Phylogenetic tree
- Include bootstrap values if mentioned in the legend.
- Resolve inconsistencies in phylogenetic method: Materials and Methods cite neighbor-joining, while the figure legend refers to maximum likelihood.
- Explain the scale bar and substitution model used.
2. 3.2. Incomplete Cross-Protection Between H1a and A Genotypes of Measles Virus
1) Data Interpretation
- No baseline or negative control data are included, making it difficult to interpret the magnitude and specificity of the immune responses.
- Please clarify the basis for “incomplete neutralization.” In Figures 2A and 2B, all animals show titers >120, yet it is unclear if these meet the known protective threshold (120 mIU/mL).
- PRNT assays have inherent variability (~threefold; doi:10.1016/j.vaccine.2007.10.046). Differences should be interpreted cautiously unless they exceed this variability.
2) Please confirm that IRB approval was obtained for the use of human serum samples from 18-month-old individuals.
3. 3.3. Patients with a history of measles vaccination reinfected with measles virus
This section may be better placed in the Introduction. If retained, please include a description of the fluorescent quantitative qPCR assay referenced in the Figure 3 legend or cite an appropriate reference.
4. 3.4. rSchwarz/FH(H1a) exhibits enhanced immunogenicity compared to the parental virus
1) Please move a higher-resolution version of Figure 4B to the Supplementary Materials.
2) Retain representative images from Figure 4C in the main manuscript and relocate remaining images to supplementary materials.
3) Please improve scale bar visibility in Figures 4B and 4C and include dimensions in the figure legends.
4) Consider incorporating epitope mapping from Supplementary Figure S2 into this section.
5. 3.5. rSchwarz/FH(H1a) Induces High Titers of Neutralizing Antibodies Against Multiple Epidemic Genotype Strains of measles virus compared to rSchwarz
1) BALB/c mice Model
- As with Section 3.2, baseline and control data are needed to accurately evaluate immune responses.
- In Figure 5B, the ~twofold difference in titers between rSchwarz and rSchwarz/FH(H1a) may fall within assay variability. Please clarify whether this was statistically significant.
2) Rhesus Monkey Model
- The study uses 10⁵ CCID₅₀ in mice vs. 10⁴.25 CCID₅₀ in monkeys. Please explain the rationale for this dose selection and whether scaling or optimization was considered.
- Figure 8B shows large standard deviations in the rSchwarz/FH group. Clarify whether this is due to vaccine variability or individual immune responses.
- Figures 8B and 8C: Animals immunized with S-191 showed very low titers of neutralizing antibodies (~10), yet the manuscript claims a fourfold increase from baseline. Please provide the baseline titer and the neutralization assay method used to support this conclusion.
6. 3.6. rSchwarz/FH(H1a) Exhibits Good Genetic Stability and immune persistence
1) Line 370: Per Figure 6A, two nucleotide changes occurred—one silent in the M gene and one missense in the L gene. Please update the text to reflect that only a single amino acid change occurred.
2) Include a detailed description of whole genome sequencing methods in the Materials and Methods section or cite an appropriate reference.
- Line 434, Figure 8B: Specify whether the data represent immunized mice or rhesus monkeys.
- Figure S2. The amino acid at position 476 of the H protein is a critical determinant for incomplete cross-protection between measles virus strains of genotypes H1a and A.
1) Please consider moving these findings to Section 3.4. rSchwarz/FH(H1a) exhibits enhanced immunogenicity compared to the parental virus as they are directly relevant to the discussion of the recombinant virus's immunogenicity.
2) Figure S2C: In the cross-neutralization experiments, please clarify why the difference in neutralizing antibody titers is more statistically significant in mice immunized with the Schwarz strain compared to those immunized with MV-1 (referring to the bars on the far right of the graph). A brief explanation would help contextualize the observed variance in response.
D. Discussion
While the study does not involve the use of a killed (inactivated) measles virus, and discussion of inactivated vaccine efficacy may not be directly applicable, it would be valuable to address the notably higher rate of breakthrough measles cases reported in China. For comparison, breakthrough infection rates in the United States and the European Region remain significantly lower (approximately 3–5%). Given that measles virus is generally considered to have a single serotype, it is important to discuss the rationale for using vaccine strains from different genotypes—such as the H1a genotype employed in this study—to enhance immunity against wild-type viruses, especially considering that H1a has not been detected in China since 2020. Is there evidence from other studies to support this genotype-based vaccine strategy? Additionally, including a brief discussion of potential contributing factors to the high rate of breakthrough infections—such as vaccine manufacturing quality, cold-chain transport conditions, immunization coverage, scheduling practices, population seroprevalence, or waning immunity—would strengthen the manuscript’s relevance and provide important context for its public health implications.
Author Response
Comments 1: Organization of the Content: The manuscript’s readability and flow could be improved by reorganizing certain sections. For example, portions of Sections 3.1 and 3.3 contain background information that may be more appropriately placed in the Introduction, where they can provide context for the experimental rationale. Additionally, Supplementary Figure S2 contains data that are central to the findings discussed in Section 3.4 and should be included in the main text to enhance clarity.
Response 1: We sincerely thank you for your valuable suggestions on improving readability of our manuscript. We fully agree with your recommendations and have partially modified content of the manuscript. The details of modified content in the revised manuscript is below: :
- The sentence from the previous Section 3.1, "The measles virus strains circulating in China are predominantly of the H1a genotype," has been placed in the Introduction (Lines 84–103).
The content previously presented in Section 3.3, "Patients with a history of measles vaccination reinfected with measles virus," has also been integrated into the Introduction (Lines 67–81).
2, The results previously shown in Supplementary Figure S2 have been incorporated into the revised Section 3.2, titled “rSchwarz/FH(H1a) exhibits improved immunogenicity, with the H protein residue 476 playing a key role in cross-protection between genotypes H1a and A measles virus.” As a result, the former Figure S2 is now Figure 5 (previously Section 3.4, “rSchwarz/FH(H1a) exhibits enhanced immunogenicity compared to the parental virus”).
Once again, we deeply appreciate your guidance to enhance the rigor and clarity of the main text.
Comments 2: Data Interpretation: The authors should clearly define statistical significance while accounting for the inherent inter-assay variability of neutralization assays.
Response 2: We appreciate the reviewer’s attention to detail. As suggested, in the Statistical Analysis in Methods section, we have clearly stated "Neutralizing antibody titers from mice, guinea pigs, NHPs, and human serum samples were presented as mean with 95% confidence intervals (95% CI). Statistical comparisons between groups for experiments to determine the percentage of survival were analyzed using Kaplan-Meier survival curves and the log-rank test. For all tests, the following notations were used to indicate significant differences between groups: *, P < 0.05; **, P < 0.01; **, P < 0.001." Regarding the micro-neutralization assay, during the initial development of the method, we strictly followed the ICH Q2(R1) guidelines to perform method validation. The validation confirmed acceptable repeatability (CV = 18%) and intermediate precision (CV = 20%), as well as a low maximum-to-minimum titer ratio (Max/Min < 2). Therefore, we conclude that the inherent inter-assay variability of this method is low and does not affect the statistical analysis results.
Comments 3: Lack of Negative Controls and Baseline Data: The absence of pre-immunization or negative control data limits the ability to assess the true magnitude of the immune response and to make meaningful comparisons of cross-neutralizing titers across different strains.
Response 3: We sincerely thank you for your thorough review and valuable comments on this manuscript. In response to the issues you raised, we have previously carried out experiments to answer this question. And clarifications have been provided as follow:
- Clarification on the negative control: During the development and validation of the neutralization assay, we systematically included a DMEM group as a negative control to exclude any potential non-specific neutralizing interference. The DMEM group showed completely negative neutralizing activity under undiluted serum conditions, with titers ≤ 0.71, which is the lower limit of detection for the assay.
- Clarification on baseline data: Regarding your concern about "missing pre-immunization data," serum samples of all animals in this study had been collected prior to initial immunization (Day 0) for neutralizing antibody titer testing. The results showed that the neutralizing activity of undiluted serum was consistently negative, with titers ≤ 0.71 , which is the lower limit of the assay. As this serves as a fundamental prerequisite for the experiment, we have added relevant descriptions in the figure legends of Figure 3, 4, 5, 6, 7 and 9 in the revised manuscript, in accordance with your suggestion.
Comments 4: Comparability to Protective Thresholds: Although the protective threshold for measles immunity is defined as 120 mIU/mL of neutralizing antibodies, the manuscript does not clearly demonstrate whether the titers elicited by the recombinant virus meet or exceed this level in the various animal models.
Response 4: We sincerely appreciate the reviewer’s insightful comments. We fully agree that the confirmed neutralizing antibody threshold of 120 mIU/mL, which is determined by PRNT (plaque reduction neutralization test) in human serum, may not be directly extrapolated to animal models due to inherent interspecies differences in immune responses. In this study, we did not attempt to define a universal protective neutralizing antibody threshold across animal models, for the following reasons: There is currently no standardized correlate of protection established for neutralizing antibodies in non-human species infected with measles virus; The micro-neutralization assay employed in this study was optimized for high-throughput screening, which may yield different readouts compared to PRNT; Furthermore, the linear relationship between PRNT and the micro-neutralization assay has not been validated in animal sera. Establishing such a correlation remains an important focus of our future work. To this end, we are currently acquiring the WHO international reference standard for measles antiserum (NIBSC 97/648), which will facilitate cross-assay calibration and improve comparability across platforms. Additionally, we acknowledge that the term “incomplete cross-protection” used in the original manuscript was imprecise and potentially misleading. Accordingly, we have revised the title of Figure 2 (now Figure 3 in the revised manuscript) from “Incomplete Cross-Protection Between H1a and A Genotypes of Measles Virus” to “Differential Immunogenicity Between Measles Virus Genotypes A and H1a: Genotype-Specific Neutralization Potency of Post-Immunization Sera”. Furthermore, all instances of “Incomplete Cross-Protection” throughout the manuscript have been replaced with “Different Cross-Protection” to more accurately reflect the observed genotype-specific neutralization patterns.
Comments 5: Lack of Challenge Studies: While the authors propose that the recombinant virus may be superior to existing vaccines, no challenge studies were conducted to confirm its protective efficacy in immunized animals following exposure to wild-type measles virus.
Response 5: We sincerely appreciate your insightful review and constructive comments. We fully agree on the importance of challenge studies in evaluating vaccine candidates. However, due to the unique characteristics of measles virus research, such experiments have not been included in the current study. To address this point, we provide the following explanations based on existing data and field consensus:
- Currently, establishing a protective challenge model for measles virus in non-human primates (NHPs) remains a significant challenge. As measles is a human-specific pathogen, intentionally inducing infection in NHPs may raise serious ethical concerns in animal experimentation.
- Despite the lack of challenge data, the following findings support our conclusion that the recombinant virus is superior to the currently available vaccine strain: during the neurovirulence assessment in monkeys, we specifically compared the neutralizing antibody responses between the existing vaccine strain and the candidate strain. The results demonstrated that the immunogenicity of the candidate strain was significantly higher than that of the current vaccine.
- We fully agree the importance of challenge studies in prior to clinical trials. However, our currently project focused on exploring a novel genotype-based live attenuated measles which could be a candidate for development of measles -mumps-rubella (MMR) vaccine. Therefore, challenge studies are essential works in the future.
Comments 6: Please spell out abbreviations upon first use. For example, spell out CCID₅₀ as cell culture infectious dose 50% when it first appears in the manuscript.
Response 6: Thank you for highlighting this important editorial oversight. We have carefully revised the manuscript to ensure that all abbreviations are spelled out in full upon first mention, followed by the abbreviated form in parentheses. Specifically, in line 192 (Materials and Methods section), "CCID₅₀" has been updated to "50% cell culture infectious dose (CCID₅₀)".
Comments 7: Please ensure consistent formatting throughout the manuscript. For instance, while most dose values are presented in standard notation, scientific notation is used in Line 187. Additionally, terminology should be uniform across figures. For example, use a single defined term such as NT titer consistently on the Y-axis rather than alternating between NT titer and NT₅₀.
Response 7: Thank you for the careful attention to formatting consistency. We have addressed the issues as follows: the original text at line 187 (revised manuscript line 253), which stated “log 4.25 CCID50”, has been corrected to “1.78×10^4 CCID50”. In addition, in the original Figure 8 (now Figure 9 in the revised manuscript), “NT50” has been changed to “NT titer” to ensure consistent terminology across all figures.
Comments 8: Vero/hSLAM cells may require a license in China due to regulations involving dual-use biological agents and the Convention on International Trade in Endangered Species. Please confirm whether appropriate permissions were obtained (Materials and Methods/ Cells and Viruses).
Response 8: Thank you for your attention to the compliance of biological materials used in our study. With the use permissions for Vero/hSLAM cells, we have fully confirmed that this research strictly adheres to China's biosafety regulations and the Convention on International Trade in Endangered Species of Wild Fauna and Flora (CITES). The Vero cells (ATCC number: CCL-81) used in our experiments were purchased from the American Type Culture Collection (ATCC) and are internationally recognized as a standardized cell line derived from non-endangered species. The Vero/hSLAM cell line, a genetically modified variant, was developed by Cyagen Biosciences Inc. (Guangdong) based on the Vero cells originally acquired from ATCC. These modified cells were solely utilized for the rescue and identification of recombinant measles virus in our study and were not involved in large-scale production or commercial applications. Additionally, we have added the following information to the Materials and Methods section 2.1: “Vero cells were purchased from the American Type Culture Collection (ATCC). The Vero/hSLAM cells were developed by Cyagen Biosciences Inc. (Guangdong), based on the Vero cells originally acquired from ATCC.”
Comments 9: Please provide complete information for all virus strains used (D8, B3, MV-1, Schwarz, S191), including strain names, GenBank accession number (if available), preparation methods, and passage numbers (Materials and Methods/ Cells and Viruses).
Response 9: Thank you for your attention to the traceability information of the virus strains used in our study. We have now provided the complete details for all virus strains and added the following text to Section 2.1 of the "Materials and Methods" in the revised manuscript: “The strains D8 and MV-1, measles virus strains isolated by the Jiangsu Provincial Center for Disease Control and Prevention (CDC) from clinical samples, was passaged twice before experimental use. The Schwarz vaccine strain, purchased from the University of Georgia in the United States, was passaged twice in our laboratory before use in experiments. S191, a national reference strain for live attenuated measles vaccine, was obtained from the National Institutes for Food and Drug Control in China. The rSchwarz/FH(B3) is a recombinant virus strain constructed by King-cell Biotech, using the Schwarz strain as a backbone but replacing the F and H protein genes which were from a genotype B3 measles virus (GenBank accession number: KY969477.1). After recombinant virus were successfully rescued, the virus was continuously passaged three times in Vero/hSLAM cells before use in experiments.”
Comments 10: Consider merging Sections 2.5 and 2.7 (Virus Titer Determination) to streamline the methodology (Materials and Methods/ Growth Characteristics of Recombinant Measles Virus).
Response 10: Thank you for your valuable suggestion on optimizing the content of the Methods section. In response, we have merged the original Sections 2.5 and 2.7 into a single Section 2.5 in the revised manuscript. The updated Section 2.5 is “Vero and CEF cells were seeded into 12-well plates at a density of 4×10^5 cells per well and incubated overnight at 37°C. When the cell confluence reached approximately 90%, the cells were washed thrice with PBS and then infected with the virus at an MOI of 0.01. After an hour incubation at 37°C, the inoculum was removed, and the cells were rinsed twice with PBS. Subsequently, the cell culture medium was replaced, and the cells were cultured at 35°C in a 5% CO2 atmosphere. Samples were collected every 24 hours for five consecutive days, and the virus samples were 10-fold serially diluted, and 100 μL of each dilution was added to 96-well plates pre-seeded with Ver/-hSLAM cells (1.4 × 10^5 cells/mL, 100 μL per well). Each dilution was tested in eight replicate wells. Following inoculation, the plates were incubated at 35°C under a 5% CO2 atmosphere for 7 days. Cytopathic effects (CPE) were observed daily using an inverted optical microscope. The CCID50 (cell culture infectious dose 50%) was determined using the Reed-Muench method based on wells with observed CPE.”
Comments 11: Clarify whether the temperature difference between virus adsorption (37 °C; Line 126) and subsequent incubation (35 °C; Line 128) is intentional (Materials and Methods/ Growth Characteristics of Recombinant Measles Virus).
Response 11: We sincerely thank the reviewer for pointing out this issue. We confirm that all steps, including virus adsorption and subsequent incubation, were consistently performed at 37 °C. The discrepancy in the description was due to a typographical error in the manuscript. We have corrected the temperature from "35 °C" to "37 °C" in lines 186, 190, and 297 of the revised manuscript.
Comments 12: Line 129: Expand on how virus titers were determined. Indicate that Vero/hSLAM cells were used (as per Line 324), and specify how cytopathic effects were assessed (e.g., crystal violet staining, neutral red, or light microscopy) (Materials and Methods/ Growth Characteristics of Recombinant Measles Virus).
Response 12: Thank you for your attention to the details of the virus titer measurement method. We have clarified in Section 2.5 of the revised manuscript that “Samples were collected every 24 hours for five consecutive days, and the virus samples were 10-fold serially diluted, and 100 μL of each dilution was added to 96-well plates pre-seeded with Ver/-hSLAM cells (1.4 × 10^5 cells/mL, 100 μL per well). Each dilution was tested in eight replicate wells. Following inoculation, the plates were incubated at 35°C under a 5% CO2 atmosphere for 7 days. Cytopathic effects (CPE) were observed daily using an inverted optical microscope. The CCID50 (cell culture infectious dose 50%) was determined using the Reed-Muench method based on wells with observed CPE.”
Comments 13: Line 133: Please describe the method used to concentrate supernatant from 500 mL to 14 mL (Materials and Methods/ Scanning Electron Microscopy).
Response 13: Thank you for your attention to the virus concentration method. We have added the details in Section 2.6 of the revised manuscript: “The virus supernatant was concentrated from 500 mL to 14 ml using the Pellicon XL Ultrafiltration Module, Biomax 100 kDa, 0.005 m² (Merck, Catalog No.: PXB100C50).”
Comments 14: Spell out ED50 on first mention and note that neutralizing antibody titers are typically reported as NT50 or nAb50 to denote 50% neutralization (Materials and Methods/ Micro-neutralization assay).
Response 14: Thank you for your important correction regarding the consistency and accuracy of terminology. We have systematically revised the manuscript according to your suggestions to ensure the correct definition and uniformity of abbreviations. Specifically, we have changed "ED50" to "NT50 (50% neutralization titer)" in Section 2.13 of the original manuscript to ensure that the description is more accurate.
Comments 15: This section provides useful background but is not central to the main findings. Consider moving it to the Introduction (Results/ The measles virus strains circulating in China are predominantly of the H1a genotype).
Response 15: Thank you for your valuable suggestion on optimizing the paper content. We fully agree that this background information is more appropriately included as part of the foundational rationale in the Introduction. Accordingly, the original Section 3.1 has been incorporated into lines 84–103 of the Introduction in the revised manuscript.
Comments 16: Phylogenetic tree should Include bootstrap values if mentioned in the legend; Resolve inconsistencies in phylogenetic method: Materials and Methods cite neighbor-joining, while the figure legend refers to maximum likelihood; Explain the scale bar and substitution model used (Results/ The measles virus strains circulating in China are predominantly of the H1a genotype).
Response 16: Thank you for your meticulous review of the phylogenetic analysis method details. We have verified and corrected the inconsistent descriptions in the manuscript. In lines 96–100 of the revised manuscript, we have added all tree-building parameters: “(A) A phylogenetic tree was constructed using MEGA7 with the Neighbor-Joining method. Bootstrap analysis was conducted with 500 replicates, and the Maximum Composite Likelihood model was applied under the nucleotide substitution setting. The scale bar of 0.01 corresponds to a genetic distance of one nucleotide substitution per 100 sites on average. 21 isolated strains of measles virus are highlighted in red color font.”
Comments 17: No baseline or negative control data are included, making it difficult to interpret the magnitude and specificity of the immune responses (Results/ Incomplete Cross-Protection Between H1a and A Genotypes of Measles Virus).
Response 17: Thank you for your important reminder regarding data integrity. We have added this content to lines 348–351 of the revised manuscript: “All animals underwent blood collection and neutralization titer testing prior to immunization. Following an original dilution of serum from all experimental animals, no viral inhibitory activity was observed in the micro-neutralization assay, as indicated by 100% cytopathic effect (CPE). According to the Reed-Muench method, the calculated theoretical titer was 0.71.”
Comments 18: Please clarify the basis for “incomplete neutralization.” In Figures 2A and 2B, all animals show titers >120, yet it is unclear if these meet the known protective threshold (120 mIU/mL) (Results/ Incomplete Cross-Protection Between H1a and A Genotypes of Measles Virus).
Response 18: Thank you for your valuable comments on the accuracy of terminology and data interpretation. We fully agree that the term “incomplete neutralization” may be misleading and could lead to misinterpretation. Accordingly, we have made the following revisions and clarifications in the revised manuscript: “The 120 mIU/mL correlate of protection was established based on human serum samples tested by the plaque reduction neutralization test (PRNT) (doi:10.1016/j.jcv.2015.06.095). However, due to inherent interspecies differences, this threshold may not be directly applicable to animal models. Moreover, the linear relationship between PRNT and the micro-neutralization assay used in this study has not yet been established. As part of our ongoing work, we are acquiring the WHO international reference standard (NIBSC 97/648) to address this gap". Given these limitations, the use of the term “incomplete cross-protection” is inappropriate. Therefore, we have revised the title of Figure 2 in the original manuscript (now Figure 3 in the revised manuscript) from “Incomplete Cross-Protection Between H1a and A Genotypes of Measles Virus” to “Differential Immunogenicity Between Measles Virus Genotypes A and H1a: Genotype-Specific Neutralization Potency of Post-Immunization Sera.”
Comments 19: PRNT assays have inherent variability (~threefold; doi:10.1016/j.vaccine.2007.10.046). Differences should be interpreted cautiously unless they exceed this variability (Results/ Incomplete Cross-Protection Between H1a and A Genotypes of Measles Virus).
Response 19: Thank you for your important reminder regarding the variability of neutralizing antibody assays. In response to the reviewer’s concern about the inherent variability of PRNT (approximately 3-fold), we would like to clarify the methodological advantages of the micro-neutralization assay used in this study. The variability associated with PRNT primarily stems from multiple procedural steps, including uneven agarose overlay, fluctuations in cell monolayer integrity, and subjective plaque counting. In contrast, the micro-neutralization assay employed in our study features an optimized design that eliminates fixation and staining steps and utilizes 96-well plate-based parallel detection, thereby significantly reducing operational variability. Furthermore, we conducted rigorous method validation in accordance with ICH Q2(R1) guidelines. The results demonstrated that the micro-neutralization assay exhibits excellent repeatability (CV = 18%) and intermediate precision (CV = 20%), both of which are superior to those typically observed for PRNT. Additionally, the ratio of maximum to minimum titer (Max/Min < 2) was substantially lower than the 3-fold threshold, further confirming the improved consistency and reliability of our assay.
Comments 20: Please confirm that IRB approval was obtained for the use of human serum samples from 18-month-old individuals (Results/ Incomplete Cross-Protection Between H1a and A Genotypes of Measles Virus).
Response 20: Thank you for highlighting this important ethical review information. We apologize for the omission and have added the relevant statement in Section 3.1 of the revised manuscript: “A total of 49 serum samples were obtained from 18-month-old infants in Changzhou, Jiangsu Province, China, one month following immunization with the measles-mumps-rubella (MMR) vaccine developed by the Beijing Institute of Life Sciences. The samples were used to determine neutralizing antibody titers specific to different measles virus genotypes. This study was approved by the Ethics Committee of Jiangsu Provincial Center for Disease Prevention and Control (NO.JSJK2020-B005-01).”
Comments 21: This section may be better placed in the Introduction. If retained, please include a description of the fluorescent quantitative qPCR assay referenced in the Figure 3 legend or cite an appropriate reference (Results/ Patients with a history of measles vaccination reinfected with measles virus).
Response 21: Thank you for your valuable suggestion on optimizing the paper content. We fully agree that this information is better suited as part of the foundational rationale in the Introduction. Accordingly, the original Section 3.3 has been incorporated into lines 67–81 of the Introduction in the revised manuscript.
Comments 22: Please move a higher-resolution version of Figure 4B to the Supplementary Materials; Retain representative images from Figure 4C in the main manuscript and relocate remaining images to supplementary materials; Please improve scale bar visibility in Figures 4B and 4C and include dimensions in the figure legends; Consider incorporating epitope mapping from Supplementary Figure S2 into this section (Results/ rSchwarz/FH(H1a) exhibits enhanced immunogenicity compared to the parental virus).
Response 22: Thank you for your detailed suggestions on improving the quality of figure presentation. We have made the following revisions to enhance the clarity and logic of data display: Images from parts of Figure 4B and 4C have been moved to the supplementary materials (Supplementary Figures S2 and S3); The scale bars in Figures 4B and 4C have been adjusted for better clarity, as seen in the revised Figure 4; Supplementary Figure S2 has been integrated into this section and can now be found in Section 3.2 of the revised manuscript.
Comments 23: As with Section 3.2, baseline and control data are needed to accurately evaluate immune responses (Results/ rSchwarz/FH(H1a) Induces High Titers of Neutralizing Antibodies Against Multiple Epidemic Genotype Strains of measles virus compared to rSchwarz/ BALB/c mice Model).
Response 23: Thank you for your important reminder regarding data integrity. We have added this content to lines 455–456 of the revised manuscript: “The neutralizing antibody titers were assessed in sera collected from all animals prior to immunization, with all samples showing a titer of 0.71 calculated by the Reed-Muench method.”
Comments 24: In Figure 5B, the ~twofold difference in titers between rSchwarz and rSchwarz/FH(H1a) may fall within assay variability. Please clarify whether this was statistically significant (Results/ rSchwarz/FH(H1a) Induces High Titers of Neutralizing Antibodies Against Multiple Epidemic Genotype Strains of measles virus compared to rSchwarz/ BALB/c mice Model).
Response 24: Thank you for your rigorous review of the neutralizing antibody titer differences. The neutralizing antibody assay used in this study has been thoroughly validated for specificity, precision and robustness. The precision of the method was assessed through six independent experiments for repeatability, and intermediate precision was evaluated by two analysts each performing three independent runs, yielding coefficients of variation (CV) of 18% and 20%, respectively. Additionally, the maximum-to-minimum titer ratio across all tests was consistently less than 2. These results demonstrate that the inherent variability of the method is low and does not obscure true inter-group differences — the systematic error is much smaller than the biological variability observed. Therefore, the difference in antibody titers between the rSchwarz and rSchwarz/FH (H1a) groups is statistically significant and biologically meaningful.
Comments 25: The study uses 10⁵ CCID₅₀ in mice vs. 10⁴.25 CCID₅₀ in monkeys. Please explain the rationale for this dose selection and whether scaling or optimization was considered (Results/ rSchwarz/FH(H1a) Induces High Titers of Neutralizing Antibodies Against Multiple Epidemic Genotype Strains of measles virus compared to rSchwarz/ Rhesus Monkey Model).
Response 25: Thank you for your attention to the dose selection in our animal experiments. The immunization doses used for the monkeys in this study were determined based on relevant guidelines and referenced research. The scientific basis is as follows: According to the WHO Technical Report Series on the Evaluation of Neurovirulence of Measles Vaccines, the viral dose for monkey neurovirulence testing should not be less than the viral content of a single dose of measles vaccine (typically 103CCID₅₀). Previous studies by the Chinese Academy of Biological Products have shown that a dose of 104 CCID₅₀ can reliably induce an immune response in monkeys without causing neurotoxicity (DOI: 10.3969/j.issn.1004-5503.2001.03.013; DOI: CNKI:SUN:ZGJM.0.2015-01-017). In this study, we used a dose of 104.25 CCID₅₀ (approximately 1.78 × 104 CCID50) to slightly increase the challenge dose, ensuring detectable virulence signals while avoiding excessive stress on the animals.
Comments 26: Figure 8B shows large standard deviations in the rSchwarz/FH group. Clarify whether this is due to vaccine variability or individual immune responses (Results/ rSchwarz/FH(H1a) Induces High Titers of Neutralizing Antibodies Against Multiple Epidemic Genotype Strains of measles virus compared to rSchwarz/ Rhesus Monkey Model).
Response 26: Thank you for your insightful observation regarding this data distribution characteristic. We have analyzed the data and confirmed that the standard deviation observed in the rSchwarz/FH group is primarily due to individual variation in immune responses, rather than vaccine variability. Our serum samples were tested in duplicate, and the results showed no remarkable errors with assay consistency. Furthermore, immunogenicity studies of rSchwarz/FH in other animal models (mice and guinea pigs) also ruled out vaccine-related variability. While both the rSchwarz/FH and S191 groups exhibited some degree of inter-individual variation, the higher overall immunogenicity of rSchwarz/FH resulted in a wider range of neutralizing antibody titers — individuals with strong immune responses produced high titers, while those with weaker responses generated lower titers. A larger proportion of animals in the rSchwarz/FH group exhibited strong responses, leading to greater dispersion in values and thus a larger standard deviation. In contrast, the S191 group showed consistently weaker immunogenicity across individuals, resulting in lower variation and a smaller standard deviation.
Comments 27: Figures 8B and 8C: Animals immunized with S-191 showed very low titers of neutralizing antibodies (~10), yet the manuscript claims a fourfold increase from baseline. Please provide the baseline titer and the neutralization assay method used to support this conclusion (Results/ rSchwarz/FH(H1a) Induces High Titers of Neutralizing Antibodies Against Multiple Epidemic Genotype Strains of measles virus compared to rSchwarz/ Rhesus Monkey Model).
Response 27: Thank you for your valuable comments and attention to details in our work. Regarding your question, we did perform baseline titer testing and used these data to assess changes in post-immunization samples. We have added the following information to the revised manuscript: in lines 532–535, we now state “The neutralizing antibody titers were assessed in sera collected from all rhesus macaques prior to immunization, with all samples showing a titer of 0.71 as determined by the Reed-Muench method. And the neutralizing titers of the immune sera from the DMEM group were also all 0.71.” and in lines 540–541, we have included “The neutralizing titer of pre-immunization sera (0.71) was used as the baseline for neutralizing titer assessment. Therefore, a neutralizing titer of ≥2.84 was considered positive.”
Comments 28: Line 370: Per Figure 6A, two nucleotide changes occurred—one silent in the M gene and one missense in the L gene. Please update the text to reflect that only a single amino acid change occurred (Results/ rSchwarz/FH(H1a) Exhibits Good Genetic Stability and immune persistence).
Response 28: Thank you for your thorough and meticulous review, which has helped us identify and correct the omissions in our text. We have made the following changes in lines 465–467 of the revised manuscript: “Compared to P0, a single amino acid mutation was detected in L proteins of measles viruses following seven consecutive serial passages.”
Comments 29: Include a detailed description of whole genome sequencing methods in the Materials and Methods section or cite an appropriate reference (Results/ rSchwarz/FH(H1a) Exhibits Good Genetic Stability and immune persistence).
Response 29: Thank you for your emphasis on methodological transparency. We have added detailed whole-genome sequencing methodology in Section 2.13 of the revised manuscript: “Viral samples of rSchwarz/FH(H1a) at passage P0, P7, P9, P12, P17 and P22 in chicken embryo fibroblast (CEF) cells were submitted to Shanghai Sequanta Technologies Co., Ltd. for next-generation sequencing (NGS)-based analysis to determine the viral genome sequences. Briefly, total nucleic acids were extracted from each sample, followed by reverse transcription to synthesize complementary DNA (cDNA). The cDNA was then subjected to poly-A tailing, adapter ligation, and PCR amplification to generate RNA sequencing libraries. Sequencing was performed using an Illumina platform. The resulting sequence data were subjected to bioinformatics assembly processes to re-construct the complete full-length genomic sequences of the virus.”
Comments 30: Line 434, Figure 8B: Specify whether the data represent immunized mice or rhesus monkeys.
Response 30: Thank you for your careful review. We apologize for the writing error and the data specified in Figure 8B refers to rhesus macaques. We have corrected this in the revised manuscript by changing “mice” to “rhesus macaques” in line 532.
Comments 31: Please consider moving these findings to Section 3.4. rSchwarz/FH(H1a) exhibits enhanced immunogenicity compared to the parental virus as they are directly relevant to the discussion of the recombinant virus's immunogenicity (Figure S2. The amino acid at position 476 of the H protein is a critical determinant for incomplete cross-protection between measles virus strains of genotypes H1a and A).
Response 31: Thank you for your valuable suggestions on optimizing the paper content. We fully agree that the revised content enhances the clarity of the internal logic of our findings. Figure S2 has now been integrated into Section 3.4, and the specific content can be found in lines 373–385 of the revised manuscript.
Comments 32: Figure S2C: In the cross-neutralization experiments, please clarify why the difference in neutralizing antibody titers is more statistically significant in mice immunized with the Schwarz strain compared to those immunized with MV-1 (referring to the bars on the far right of the graph). A brief explanation would help contextualize the observed variance in response.
Response 32: Thank you for your insightful observation regarding this data characteristic. After thorough verification, we confirm that the significance statistics are correct. The significant difference between the Schwarz strain and MV-1 immunization groups is primarily due to differences in data variability (dispersion) within each group.
Comments 33: While the study does not involve the use of a killed (inactivated) measles virus, and discussion of inactivated vaccine efficacy may not be directly applicable, it would be valuable to address the notably higher rate of breakthrough measles cases reported in China. For comparison, breakthrough infection rates in the United States and the European Region remain significantly lower (approximately 3–5%). Given that measles virus is generally considered to have a single serotype, it is important to discuss the rationale for using vaccine strains from different genotypes—such as the H1a genotype employed in this study—to enhance immunity against wild-type viruses, especially considering that H1a has not been detected in China since 2020. Is there evidence from other studies to support this genotype-based vaccine strategy? Additionally, including a brief discussion of potential contributing factors to the high rate of breakthrough infections—such as vaccine manufacturing quality, cold-chain transport conditions, immunization coverage, scheduling practices, population seroprevalence, or waning immunity—would strengthen the manuscript’s relevance and provide important context for its public health implications.
Response 33: Thank you for your profound insight into the public health implications of our research. Regarding your question about whether other studies support this genotype-based vaccine strategy. We have finished a lot of work before 2020 when the H1a genotype has not been detected in China since then. And B3 and D8 genotypes of measles virus become major prevalent on global since 2020. Therefore, we tried to used B3 and D8 genotypes of measles virus as challenge viruses immunization studies in our candidate vaccine, and the results showed candidate vaccine can induce highly effective cross-neutralizing antibodies against multiple circulating strains such as B3 and D8.
Moreover, the H1a genotype was the predominant circulating strain in China from 2010 to 2018. Considering the longevity of immune memory, coverage of this genotype would enhance cross-protection in previously immunized populations.
Additionally, in response to your suggestion, we have added an analysis of potential factors contributing to high breakthrough infection rates in the discussion section. Specifically, lines 560–566 now include: “Several factors could contribute to high breakthrough infection rates, including issues related to vaccine storage, transportation, administration techniques, waning immunity in the vaccinated population, and insufficient immunization coverage. However, another critical aspect to consider is the antigenic drift between the circulating strains and the vaccine strains used. The divergence in antigenicity between prevalent virus strains and the vaccine strains can significantly impact vaccine efficacy and may explain some of the observed breakthrough infections.”
Reviewer 2 Report
Comments and Suggestions for Authors
In this study, experiments were conducted to evaluate the reassortant of the Schwarz measles virus vaccine strain, widely used in the world, with the wild measles virus genotype H1, endemic for China.
During the in vivo experiments, the following parameters were evaluated using relevant animal models:
- evaluation of the immunogenicity of the new reassortant strain in three animal species (balb/c mice, guinea pigs, rhesus macaques)
- infection test (punch test) of genetically modified IFNα/βR-/- (A129) mice (deficient in the production of alpha and beta interferons)
- genetic stability of rSchwarz/FH (H1a) as a vaccine candidate.
- evaluation of the neurovirulence of the reassortant strain in rhesus macaques, etc.
At the same time, there is no description of a number of parameters required for the correct evaluation of the results of preclinical studies, such as
- acute toxicity
- chronic toxicity
- local irritant and allergenic effects.
An important priority issue remains the safety characterization of rSchwarz/FH (H1a) as a vaccine candidate when deciding whether to proceed to human studies. Despite the fact that experimental data have shown the absence of neurovirulence of the recombinant virus, one cannot help but take into account that in addition to the main genes P and M, which determine the virulence of the measles virus, the genes F and H also exhibit virulence characteristics, so their inclusion in the recombinant vaccine can lead to an imbalance in the parent (natural) strain, which has long been used in vaccine preparations for more than 50 years. That is, the question arises: can the reactogenicity of the vaccine candidate increase, affecting the functions of other organs and systems? After all, the vaccine must be safe first of all.
In the case of using a candidate vaccine for immunizing children, regulators in a number of countries may require toxicological studies on immature animals - are there such studies?
The presented results are promising, confirming the broad cross-reactivity and immunogenicity of the new vaccine strain, and disclosing the answers to the questions posed will improve the quality of the presented work.
Author Response
Comments 1: During the in vivo experiments, the following parameters were evaluated using relevant animal models:
- evaluation of the immunogenicity of the new reassortant strain in three animal species (balb/c mice, guinea pigs, rhesus macaques)
- infection test (punch test) of genetically modified IFNα/βR-/- (A129) mice (deficient in the production of alpha and beta interferons)
- genetic stability of rSchwarz/FH (H1a) as a vaccine candidate.
- evaluation of the neurovirulence of the reassortant strain in rhesus macaques, etc.
At the same time, there is no description of a number of parameters required for the correct evaluation of the results of preclinical studies, such as
- acute toxicity
- chronic toxicity
- local irritant and allergenic effects.
An important priority issue remains the safety characterization of rSchwarz/FH (H1a) as a vaccine candidate when deciding whether to proceed to human studies. Despite the fact that experimental data have shown the absence of neurovirulence of the recombinant virus, one cannot help but take into account that in addition to the main genes P and M, which determine the virulence of the measles virus, the genes F and H also exhibit virulence characteristics, so their inclusion in the recombinant vaccine can lead to an imbalance in the parent (natural) strain, which has long been used in vaccine preparations for more than 50 years. That is, the question arises: can the reactogenicity of the vaccine candidate increase, affecting the functions of other organs and systems? After all, the vaccine must be safe first of all.
In the case of using a candidate vaccine for immunizing children, regulators in a number of countries may require toxicological studies on immature animals - are there such studies?
The presented results are promising, confirming the broad cross-reactivity and immunogenicity of the new vaccine strain, and disclosing the answers to the questions posed will improve the quality of the presented work.
Response 1: Thank you for your insightful comments on the preclinical evaluation parameters. We fully agree that assessing acute toxicity, chronic toxicity, local irritation, and allergic potential is essential for establishing the clinical relevance of our study. The ultimate goal of developing the new genotype measles virus vaccine is to incorporate it into a novel trivalent live-attenuated measles-mumps-rubella (MMR) vaccine, which we have now advanced to preclinical safety evaluation. In accordance with the Good Laboratory Practice (GLP) guidelines outlined in China’s National Medical Products Administration (NMPA) 2017 Guidelines for Nonclinical Safety Studies of Pharmaceuticals and the U.S. FDA's Guideline on Nonclinical Safety Evaluation of Preventive Vaccines (21 CFR Part 58), we have conducted comprehensive safety assessments, including acute and repeated-dose chronic toxicity studies in rats, as well as local irritation and active allergicity studies in guinea pigs, using both high and low-dose regimens. No adverse reactions or signs of toxicity were observed in any of the experimental animals during these studies. The related safety data are currently being prepared for publication in a separate manuscript.
Reviewer 3 Report
Comments and Suggestions for Authors
The authors developed recombinant measles virus based on Schwarz strain. The authors replaced F and /or H gene by H1a strain. They characterized generated rSchwarz/FH(H1a) virus in vitro and in vivo in mice, guinea pigs and monkeys. They have shown good antigenicity and safety of rSchwarz/FH(H1a) in mice guinea pigs as well as monkeys. I recommend to accept this manuscript after minor modifications. Specific comments follow.
Major points:
- Line 89: Please explain or add reference for “rAAV-hsyn-slam virus”.
- Line 176: Please explain “vg”.
- Figure 1: I don’t understand the color code. While 21 MV were isolated (line 245) Figure 1B shows only 11 H1a strains. What do you want to show?
- Figure 2: Please make it clear which is vaccine and which is target virus.
- Line 281: Please include human samples details, ethics statement and MMR vaccine used in M&M section.
Minor points:
- Figure 1: (A) is missing in figure legend.
- Figure 2: lease delete “Ten (line 275)” as it is confusing.
- Figure 3: “2023” should be “2024 (line292)”.
- Figures 1,2,5 and 6: Please replace “gennotype’ by “genotype”.
- Figure S2A: “477” should be “476” and include references describing epitopes. I think 4 amino acids cannot be an epitope.
Author Response
Comments 1: Line 89: Please explain or add reference for “rAAV-hsyn-slam virus”.
Response 1: Thank you for your attention to the use of "rAAV-hsyn-SLAM virus" in the manuscript. We have added the following text in lines 606-619 of the Discussion section in the revised manuscript: “The choice of animal model was based on commonly used transgenic mouse models for studying measles virus pathogenicity, including CD46-overexpressing transgenic mice [43], SLAM-overexpressing transgenic mice [44], and IFNα/βR-/- mice [45]. When IFNα/βR-/- mice immunized with Schwarz vaccine strain were challenged with measles virus, viral replication could be detected from samples at the injection site. Additionally, attenuated strain infection induced cells to produce more interferon, which significantly inhibits the replication of wild-type strains. Given that SLAM serves as a common receptor for both wild-type and vaccine strains of measles virus, we selected IFNα/βR-/- mice expressing the SLAM receptor to compare the virulence of measles virus strains. However, constructing stable SLAM-overexpressing IFNα/βR-/- mice is time-consuming and costly. So, we consideed the high and stable expression of exogenous proteins by AAV vectors by injecting rAAV-hsyn-SLAM virus intra-cranially into IFNα/βR-/- mice to establish a model capable of stably expressing the SLAM receptor in the brain.”
Comments 2: Line 176: Please explain “vg”.
Response 2: Thank you for your attention to the abbreviation "vg" in the manuscript. "vg" stands for “viral genomes”, which refers to the number of viral genome copies per unit volume (e.g., per milliliter, mL), typically designated as vg/mL. In this study, we used qPCR to directly measure the concentration of viral DNA. We have now added the clarification “vg (viral genomes)” in line 241 of the revised manuscript.
Comments 3: Figure 1: I don’t understand the color code. While 21 MV were isolated (line 245). Figure 1B shows only 11 H1a strains. What do you want to show?
Response 3: Thank you very much for your attention to Figure 1. In Figure 1B, the colors represent the quantity scale, indicating the number of viral strains isolated per province. Three provinces are shown in red, meaning that five viral strains were isolated from each of these provinces, totaling 15 strains. Therefore, the total number of strains represented in Figure 1B is 21.
Comments 4: Figure 2: Please make it clear which is vaccine and which is target virus.
Response 4: Thank you for your careful review of Figure 2. We acknowledge that the vaccine strains and target viruses in Figure 2 could indeed be confusing. To address this, we have added clarifying annotations directly in the figure. Specifically, these changes can be found in the revised manuscript’s Figure 3.
Comments 5: Line 281: Please include human samples details, ethics statement and MMR vaccine used in M&M section.
Response 5: Thank you for highlighting this important ethical review information. We apologize for the omission and have added the relevant statement in lines 330–336 of Section 3.1 in the revised manuscript: “A total of 49 serum samples from 18-month-old infants in Changzhou city of Jiangsu Province in China after a month when the children were immunized with the measles-mumps-rubella (MMR) vaccine developed by the Beijing Institute of Life Sciences. The samples were analyzed to determine neutralizing antibody titers specific to different measles virus genotypes. This study was approved by the Ethics Committee of Jiangsu Provincial Center for Disease Prevention and Control (NO.JSJK2020-B005-01).”
Comments 6: Figure 1: (A) is missing in figure legend.
Response 6: We sincerely apologize for the typographical error. We have added the missing label "(A)" in the figure legend at line 96 of the revised manuscript.
Comments 7: Figure 2: lease delete “Ten (line 275)” as it is confusing.
Response 7: We sincerely apologize for the typographical error. In the revised manuscript, We have removed the word "Ten" from line 275 of the original text.
Comments 8: Figure 3: “2023” should be “2024 (line292)”.
Response 8: We sincerely apologize for the typographical errors. Upon reviewing the entire manuscript, we identified three instances of the same mistake. These have been corrected in line 70, 76 and 558 of the revised manuscript.
Comments 9: Figures 1,2,5 and 6: Please replace “gennotype’ by “genotype”.
Response 9: We sincerely apologize for the typographical errors. In the revised manuscript, the misspelled word "gennotype" has been corrected to "genotype" in Figure 2, 3, 6 and 7.
Comments 10: Figure S2A: “477” should be “476” and include references describing epitopes. I think 4 amino acids cannot be an epitope.
Response 10: We sincerely apologize for the typographical error. In Figure S2A, the sequence "473-477" is correct, as the specific epitope is documented in the literature (doi:10.3390/v8080216). However, Figure S2A should display five amino acids, and it was missing the 477th amino acid. In the revised manuscript, specifically in Figure S2A (now Figure 5), we have corrected "IPRF" to "IPRFK" and "IPRL" to "IPRLK" to include the missing 477th amino acid.
Reviewer 4 Report
Comments and Suggestions for Authors
This paper describes the construction of a recombinant measles virus using Schwartz strain as a backbone. Into this were incorporated the immunogenic genes from the H1a genotype of measles virus isolated from China. The rationale is that H1a genotypes is most commonly detected in China. The authors claim that a recombinant vaccine with H1a surface protein might induce better protection against this particular genotype. Indeed, they show that mice immunized with the H1a recombinant virus develop higher levels of neutralizing antibody against H1a than those immunized with the Schwartz strain (and vice versa, Schwartz strain induces higher level of homologous antibodies).
The report originates from a private company, King-Cell Biotechnology, and there are no academic coauthors on the paper. Therefore, it may be like a company advertisement. On the other hand, the paper is well and fluently written, although the tone is over-optimistic. It is not clear what the real need for a H1a recombinant measles vaccine might be. So far, the Schwartz and Moraten strain vaccines have been extremely successful in elimination of measles worldwide.
The title promises too much. This is a new recombinant measles virus but hardly a vaccine. The material used in immunization experiments is just infected tissue culture fluid. The title should be more specific: rather than saying “novel” it should say H1a recombinant. Instead of vaccine it should say, at most, “candidate” vaccine. Evaluation should be defined as “preclinical”.
The source of Schwartz strain should be disclosed. It now says only “King-Cell Biotechnology”, but where did King-Cell obtain it from?
Author Response
Comments 1: The report originates from a private company, King-Cell Biotechnology, and there are no academic coauthors on the paper. Therefore, it may be like a company advertisement. On the other hand, the paper is well and fluently written, although the tone is over-optimistic. It is not clear what the real need for a H1a recombinant measles vaccine might be. So far, the Schwartz and Moraten strain vaccines have been extremely successful in elimination of measles worldwide.
Response 1: Thank you for your meticulous review of our manuscript and for your valuable insights regarding the independence of our research and the rationale behind the need for new vaccine development. Below, we provide further details on the genuine need for developing an H1a genotype recombinant measles vaccine, as discussed in both the Introduction and Discussion sections of our manuscript:
- Despite the high efficacy of existing live attenuated measles vaccines, all currently available vaccines are based on the A genotype. However, there are no longer any circulating A genotype measles virus. New genotype vaccines can better mimic the natural infection process, thereby inducing stronger and more durable immune responses.
- The existing A genotype measles vaccines exhibit varying levels of cross-protection against other circulating genotypes such as H1a, B3, and D8 (doi:10.1093/femspd/ftw089; doi:10.3390/v15112243). This indicates that the currently used A genotype measles vaccine may not provide adequate protection against all circulating measles virus strains.
- The study on measles infection in Jiangsu Province of China (Figure 1 of the revised manuscript), showed that between 2018 and 2024, at least 44.44% of confirmed measles cases occurred in individuals who had received at least one dose of the A genotype measles vaccine. This suggests that the current vaccines may be insufficiently effective against certain circulating strains.
Given these points, developing new genotype live attenuated measles vaccines is essential to overcome immune escape caused by antigenic drift and to enhance vaccine effectiveness against various measles virus genotypes. In this study, rSchwarz/FH(H1a) candidate vaccine has demonstrated the its ability to induce robust immune responses against the prevalent H1a, B3 and D8 genotypes of measles virus.
Comments 2: The title promises too much. This is a new recombinant measles virus but hardly a vaccine. The material used in immunization experiments is just infected tissue culture fluid. The title should be more specific: rather than saying “novel” it should say H1a recombinant. Instead of vaccine it should say, at most, “candidate” vaccine. Evaluation should be defined as “preclinical”.
Response 2: Thank you for your important correction regarding the precision of the title. We fully agree with your suggestion and have made the following substantial revisions to the title and related descriptions. The title has now been changed from “Construction and Evaluation of a Novel Attenuated Live Measles Virus Vaccine” to “Construction and Preclinical Evaluation of a Recombinant Attenuated Measles Vaccine Candidate of the H1a Genotype.”
Comments 3: The source of Schwartz strain should be disclosed. It now says only “King-Cell Biotechnology”, but where did King-Cell obtain it from?
Response 3: Thank you for your attention to the strain origin. We have added the relevant information “The Schwarz vaccine strain, purchased from the University of Georgia in the United States, was passaged twice in our laboratory before use in experiments.” in Section 2.1 (Materials and Methods) of the revised manuscript.
Round 2
Reviewer 1 Report
Comments and Suggestions for Authors
- The Vero/hSLAM cell line was developed by Ono et al. (DOI: 10.1128/JVI.75.9.4399-4401.2001). If use of this cell line results in a scientific publication, it should be cited as: Vero/hSLAM (ECACC 04091501). Unless the cell line was independently recreated by Cyagen Biosciences Inc. (Guangdong, China), please follow the appropriate citation guidance as outlined here: https://www.culturecollections.org.uk/nop/product/verohslam.
- The estimated titer of 0.71 should not be used in calculating a four-fold rise (2.84). Titers determined by the Reed-Muench method are only valid if the endpoint falls within the range of the serial dilutions. Specifically, if the undiluted serum fails to reduce the signal from the input virus by at least 50%, the titer should be reported as "<2"—reflecting the 1:2 dilution from mixing equal volumes of serum and virus—rather than assigning a numeric value below 2.
- Figure 1 and Figure 2 may not be necessary, as the relevant information is already well-documented in existing publications. For example, genotype H1 has been endemic in China since the late 1990s, and numerous studies have reported on the progress of measles elimination, including molecular surveillance over this period. Additionally, a study of measles cases in Tianjin, China, reported that over 25% of cases occurred in individuals who had received at least one dose of the measles vaccine prior to infection (DOI: 10.1016/j.vaccine.2019.05.005).
- Figure 1 – The phylogenetic tree does not display bootstrap values. Please include these to support the reliability of the branching.
- Line 307 – Please specify which Illumina platform was used for next-generation sequencing (e.g., Illumina MiSeq, HiSeq, or another model).
- Lines 371, 554, and 556 – While replacing the word "incomplete" with "different" improves clarity, indicating the fold difference would further enhance scientific accuracy and precision.
- Lines 565–567 – Please provide appropriate references to support the statement: “The divergence in antigenicity between prevalent virus strains and the vaccine strains can significantly impact vaccine efficacy and may explain some of the observed breakthrough infections.” References from studies on influenza virus and SARS-CoV-2 may be useful in substantiating this point.
- MV-1 strain – Please provide the name of MV-1 using the standard WHO nomenclature for wild-type measles virus, as outlined in: Measles virus nomenclature update: 2012. Wkly Epidemiol Rec. 2012 Mar 2;87(9):73–81.
Author Response
Comments 1: The Vero/hSLAM cell line was developed by Ono et al. (DOI: 10.1128/JVI.75.9.4399-4401.2001). If use of this cell line results in a scientific publication, it should be cited as: Vero/hSLAM (ECACC 04091501). Unless the cell line was independently recreated by Cyagen Biosciences Inc. (Guangdong, China), please follow the appropriate citation guidance as outlined here: https://www.culturecollections.org.uk/nop/product/verohslam.
Response 1: Thank you for highlighting the importance of correctly citing the Vero/hSLAM cell line used in our study. As suggested, we have added the citation for the Vero/hSLAM cells (ECACC 04091501) at line 88 of our revised manuscript. The revised sentence now reads: "Vero cells were purchased from the American Type Culture Collection (ATCC). The Vero/hSLAM cells (ECACC 04091501) were developed by Cyagen Biosciences Inc. (Guangdong), based on the Vero cells originally acquired from ATCC."
Comments 2: The estimated titer of 0.71 should not be used in calculating a four-fold rise (2.84). Titers determined by the Reed-Muench method are only valid if the endpoint falls within the range of the serial dilutions. Specifically, if the undiluted serum fails to reduce the signal from the input virus by at least 50%, the titer should be reported as "<2"—reflecting the 1:2 dilution from mixing equal volumes of serum and virus—rather than assigning a numeric value below 2.
Response 2: Thank you for your comment. We failed to consider this matter with sufficient rigor. Following your advice, we have revised the pre-immunization serum titer from 0.71 to <2. These changes have been made in the revised manuscript at lines 311,363,384,414,452,491 and 492.Additionally, based on the definition of seroconversion from PMID: 2081639, which states that "seroconversion is defined as either a conversion from negative to positive or at least a two-fold increase in titer," we define a titer >2 as positive. These changes have been made in the revised manuscript at lines 495-496.
Comments 3: Figure 1 and Figure 2 may not be necessary, as the relevant information is already well-documented in existing publications. For example, genotype H1 has been endemic in China since the late 1990s, and numerous studies have reported on the progress of measles elimination, including molecular surveillance over this period. Additionally, a study of measles cases in Tianjin, China, reported that over 25% of cases occurred in individuals who had received at least one dose of the measles vaccine prior to infection (DOI: 10.1016/j.vaccine.2019.05.005).
Response 3: Thank you for your thoughtful suggestion regarding Figures 1 and 2. We agree with your assessment. We have removed both Figure 1 and Figure 2 from the manuscript. Additionally, we have incorporated the relevant finding from the Tianjin study into the revised Discussion section at lines 512-514, now stating: "A study of measles cases in Tianjin, China, reported that more than 25% of cases occurred in individuals who had received at least one dose of the measles vaccine prior to infection."
Comments 4: Figure 1 – The phylogenetic tree does not display bootstrap values. Please include these to support the reliability of the branching.
Response 4: Thank you for your comment regarding Figure 1. In response to your feedback, we have removed Figure 1 from the revised manuscript.
Comments 5: Line 307 – Please specify which Illumina platform was used for next-generation sequencing (e.g., Illumina MiSeq, HiSeq, or another model).
Response 5: Thank you for your comment on specifying the Illumina platform used for next-generation sequencing. In response to your feedback, we have added the following information at line 267 of the revised manuscript: "Sequencing was performed using an Illumina Novaseq 6000 PE 150 platform."
Comments 6: Lines 371, 554, and 556 – While replacing the word "incomplete" with "different" improves clarity, indicating the fold difference would further enhance scientific accuracy and precision.
Response 6: Thank you for your insightful comment. Your suggestion has indeed improved the clarity and scientific accuracy of our manuscript.
Comments 7: Lines 565–567 – Please provide appropriate references to support the statement: “The divergence in antigenicity between prevalent virus strains and the vaccine strains can significantly impact vaccine efficacy and may explain some of the observed breakthrough infections.” References from studies on influenza virus and SARS-CoV-2 may be useful in substantiating this point.
Response 7: Thank you for your valuable suggestion regarding the need for references to support the statement on antigenicity divergence and its impact on vaccine efficacy. We have added appropriate references at line 521 of the revised manuscript to substantiate this point.
Comments 8: MV-1 strain – Please provide the name of MV-1 using the standard WHO nomenclature for wild-type measles virus, as outlined in: Measles virus nomenclature update: 2012. Wkly Epidemiol Rec. 2012 Mar 2;87(9):73–81.
Response 8: Thank you for your comment regarding the nomenclature of the MV-1 strain. In accordance with the standard WHO nomenclature for wild-type measles virus as outlined in "Measles virus nomenclature update: 2012. Wkly Epidemiol Rec. 2012 Mar 2;87(9):73–81", we have shown the standard WHO nomenclature for wild-type measles virus MV-1 strain as "MVs/Jiangsu.CHN/38.16/1[H1a]" at lines 14-16 in the revised manuscript, as follows: "Based on a newly isolated wild-type measles virus strain (genotype H1a), designated MVs/Jiangsu.CHN/38.16/1[H1a] (MV-1), this study aims to develop and evaluate a novel recombinant measles virus vaccine candidate."